# Demonstrating the potential of yttrium-doped barium zirconate electrolyte for high-performance fuel cells

Kiho Bae[1,2], Dong Young Jang[1], Hyung Jong Choi[1], Donghwan Kim[1,2], Jongsup Hong[2], Byung-Kook Kim[2], Jong-Ho Lee[2,3], Ji-Won Son[2,3] & Joon Hyung Shim[1]

In reducing the high operating temperatures ($\geq 800\,^\circ C$) of solid-oxide fuel cells, use of protonic ceramics as an alternative electrolyte material is attractive due to their high conductivity and low activation energy in a low-temperature regime ($\leq 600\,^\circ C$). Among many protonic ceramics, yttrium-doped barium zirconate has attracted attention due to its excellent chemical stability, which is the main issue in protonic-ceramic fuel cells. However, poor sinterability of yttrium-doped barium zirconate discourages its fabrication as a thin-film electrolyte and integration on porous anode supports, both of which are essential to achieve high performance. Here we fabricate a protonic-ceramic fuel cell using a thin-film-deposited yttrium-doped barium zirconate electrolyte with no impeding grain boundaries owing to the columnar structure tightly integrated with nanogranular cathode and nanoporous anode supports, which to the best of our knowledge exhibits a record high-power output of up to an order of magnitude higher than those of other reported barium zirconate-based fuel cells.

[1] School of Mechanical Engineering, Korea University, Anam-ro 145, Seongbuk-gu, Seoul 02841, Republic of Korea. [2] High-Temperature Energy Materials Research Center, Korea Institute of Science and Technology (KIST), 5, Hwarang-ro 14-gil, Seongbuk-gu, Seoul 02792, Republic of Korea. [3] Nanomaterials Science and Engineering, Korea University of Science and Technology (UST), KIST Campus, Seoul 02792, Republic of Korea. Correspondence and requests for materials should be addressed to J.-W.S. (email: jwson@kist.re.kr) or to J.H.S. (email: shimm@korea.ac.kr).

Proton conduction in several doped perovskite oxides has opened new opportunities to use ceramic electrolytes for protonic devices, such as gas sensors, steam electrolyzers, and protonic-ceramic fuel cells (PCFCs)[1–5]. Among these, PCFCs have attracted attention because of the possibility of reducing the high operation temperature of conventional ceramic fuel cells (solid-oxide fuel cells, SOFCs, operate at typically 800–1,000 °C) to <600 °C while retaining high ionic conductivity at the low temperatures (LTs) with a significantly low activation energy (<0.5 eV)[4–7]. Since the high operating temperature is considered as a main reason for fast degradation and high cost of SOFCs, PCFCs are expected to be a potent alternative to SOFCs.

In spite of the advantages in LTs, many protonic ceramics (PCs) suffer from poor chemical stability under $H_2O$ or $CO_2$ atmosphere, which deteriorates the long-term stability of PCFCs[8–10]. In this regard, yttrium-doped barium zirconate (BZY) has been considered as an attractive electrolyte material for PCFCs due to its excellent chemical stability[6,7] as well as high bulk ionic conductivity[11–14]. This excellent chemical stability of BZY against carbon contamination was also confirmed in our preliminary experiments as described in Supplementary Figs 1 and 2. However, PCFCs so far developed with BZY electrolytes following the conventional fabrication process of SOFCs have demonstrated unsatisfactory performance (blue box in Fig. 1). The reported poor performance of BZY-PCFCs is mainly due to the high ohmic resistance of the electrolyte. One probable contributor is the highly resistive grain boundaries of BZY in proton conduction, resulting in large ohmic resistance and low-power outputs of the PCFC[15,16]. Hence, minimization or ideally elimination of the grain boundaries in the electrolytes can be beneficial during the cell fabrication of BZY-PCFCs to achieve high performance at LTs. However, poor sinterability of the BZY material requiring for a high sintering temperature (~1,700 °C) for sufficient grain growth[17,18] has discouraged successful synthesis of highly conductive dense thin-film BZY membrane. As a way to promote grain growth of BZY without high sintering temperature, the addition of sintering aids have been suggested[19,20], but the consequent conductivity reduction nullifies the merit of using BZY for replacing conventional oxygen-ion-conducting oxides. Solid-state reactive sintering, where material synthesis and sintering are carried out simultaneously using nano-size precursors, has enabled the growth of relatively large BZY grains and effectively reduced grain-boundary resistance[14,21]. However, a fuel cell having a BZY electrolyte with such large grain sizes (~1 μm) has not been reported yet to the best of our knowledge.

The most straightforward approach to lowering the ohmic resistance of the BZY electrolyte is to reduce its thickness while eliminating the impeding grain boundaries. There have been recent successes in high-conductivity measurements from thin-film-deposited BZY[12–22], confirming that fabrication of a highly conductive BZY electrolyte is possible as long as one retains the reduced thickness as well as no grain boundaries. Indeed, PCFCs with thin-film BZY electrolytes have been successfully developed using the free-standing membrane-electrode assemblies (MEAs), and demonstrated reasonably high-power outputs at the reduced temperatures below 450 °C (green box in Fig. 1). However, poor mechanical stability and limited effective areas of the free-standing MEA-based PCFCs prevent those to function as a practical device[23,24]. Here we propose use of a 'multi-scale' anode to grow thin and dense BZY membrane atop, and report the successful fabrication of a well-integrated BZY electrolyte with columnar-grain-structure being free of grain-boundary across the film. As a result, our fuel cells have marked the topmost fuel cell performance among those of the reported BZY-based PCFCs (red data points in Fig. 1). We expect that our approach may provide a potential framework to develop highly-performing PCFCs working at LTs.

## Results

**Thin-film BZY PCFC with multi-scale anode structure.** To achieve the desired structural characteristics of the BZY membrane, that is, a thin thickness and columnar microstructure while keeping the gas tightness, in the anode-supported cell configuration, the surface condition of the anode is crucial. In the case of free-standing PCFCs, fabrication of impermeable ultra-thin BZY electrolytes with thicknesses of ~100 nm was possible, because the perfectly flat and dense surfaces were provided for the thin-film deposition by the underlying substrates, single-crystal silicon (Si) wafers[25,26]. However, depositing a thin and dense electrolyte over powder-processed anode supports with micron-scale pores is substantially challenging[27], because pinholes are generated due to the selective nucleation of the film at the edges of pores[28] and incomplete coverage of the electrolyte layer is inevitable. Hence, an optimal anode structure with high-quality surface suitable to thin-film deposition is essentially required to realize high-performance thin BZY electrolyte-based PCFCs.

In this regard, multi-scale anode structure is proposed in the work, as presented in Fig. 2. The multi-scale anode structure contains nanostructure anode surface layer (nano anode functional layer, nano-AFL) over the conventional powder-processed anode body consisting of an AFL with micron-size grains (micron-AFL) and anode support. The nano-AFL is formed by the thin film deposition, in this case by pulsed laser deposition (PLD). The insertion of the nano-AFL on porous electrode supports has significantly improved the integrity of the thin electrolyte and enhanced fuel cell performances in SOFCs[29–33]. The main reasons for the improvement are (i) reduced number of defects and roughness of the anode surface, which is preferable for dense electrolyte film growth[28,30]; (ii) increased the triple phase boundary length with smaller electrode grains[31–34]; and (iii) reduced interfacial resistances with more contact area between the electrolyte and the electrode[33,35].

To obtain fully integrated and reproducible microstructure of PCFCs based on thin BZY electrolytes, however, is much more difficult in comparison with the cases of SOFCs, because poor sinterability of BZY also affects the properties of the deposited films. The poor sinterability of BZY leads to retarded

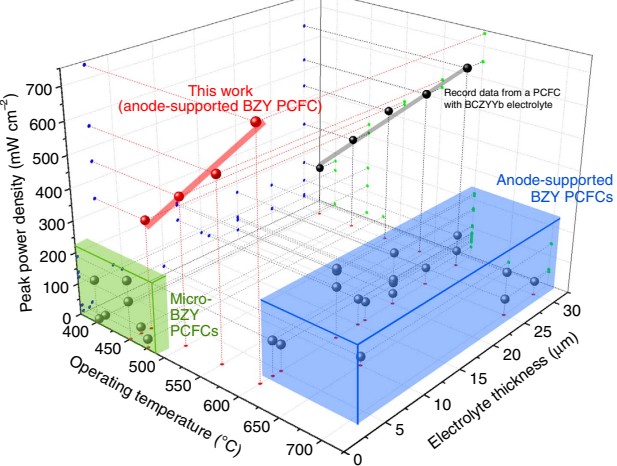

**Figure 1 | Performance comparison of acceptor-doped barium zirconate-based PCFCs.** Performance comparison of barium zirconate-based PCFCs reported in the literatures (referred to Supplementary Table 1) with the record data previously reported from a PCFC with $BaCe_{0.7}Zr_{0.1}Y_{0.1}Yb_{0.1}O_{3-\delta}$ (BCZYYb) electrolyte[39].

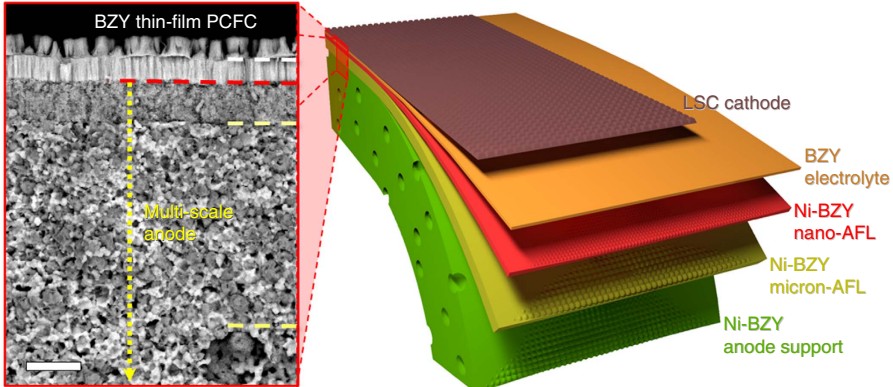

**Figure 2 | Structure configuration of the proposed BZY-PCFC.** A schematic image of the proposed configuration of anode-supported PCFCs with thin-film BZY electrolytes along with a cross-sectional SEM image of the actually fabricated PCFC in the work. Scale bar, 5 μm.

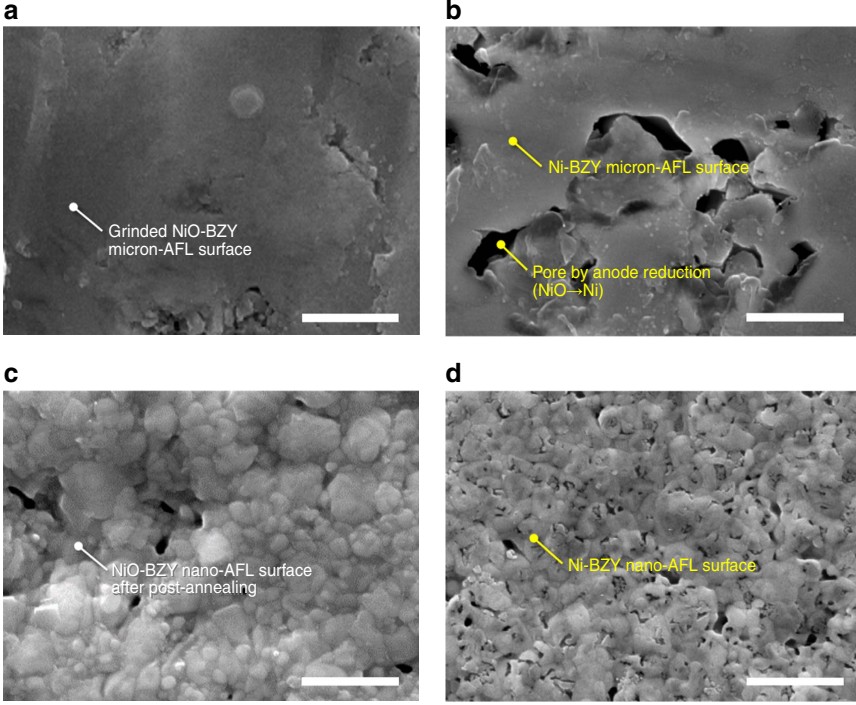

**Figure 3 | Surface morphologies of micron- and nano-ALFs.** (**a**,**b**) SEM images of the micron-AFL surface fabricated by tape-casting and sintering at 1,450 °C for 4 h after surface grinding to remove excessive grown NiO particles (**a**) and then, after anode reduction at 650 °C for 10 h under flowing of 4% $H_2$ balanced with Ar (**b**). (**c**,**d**) SEM images of the nano-AFL surface fabricated by pulsed laser deposition and post annealing at 1,300 °C for 4 h (**c**) and then, after the anode reduction (**d**). Scale bars, 1 μm.

densification in thin-film deposited and post-annealed NiO–BZY and poor interface adhesion with the anode support. Through a meticulous optimization of the multi-scale anode fabrication, we succeeded in obtaining a structurally stable and thin BZY electrolyte, as presented in the scanning electron microscopy (SEM) images in Fig. 2. More details of microstructure of the optimized PCFC are in Supplementary Fig. 3. Highly dense BZY electrolyte with a composition of $BaZr_{0.85}Y_{0.15}O_{3-\delta}$ deposited on multi-scale Ni–BZY anode with different grain and pore sizes are clearly observed. Discussion of the optimization process will be followed in the next session.

**Optimization of the BZY-PCFC fabrication.** The surface layer of the NiO–BZY anode support, micron-AFL, is formed by the tape casting, and sintered at high temperature of 1,450 °C. Due to this high-temperature sintering, the surface roughness of the micron-AFL aggravates due to the protrusion of overly grown NiO grains exhibiting BZY grain size of ∼0.5 μm or less and NiO grain size of ∼2 μm in the sintered body. Therefore, brief surface grinding was carried out and surface morphology of the micron-AFL after that is shown in Fig. 3a. After reduction of micron-scale NiO to Ni, micron-size pores are generated in micron-AFL, as shown in Fig. 3b. The large pore generation causes huge stress at the interface between anode and electrolyte and damages physical stability of the thin electrolyte floating over the pores.

To find an optimal surface morphology of the anode to sustain the thin BZY electrolyte, numerous microstructural factors, such as the grain and pore sizes, density of the surface after the post annealing, suppression of the Ni agglomeration and pore generation while the reduction, are considered for the fabrication of NiO–BZY nano-AFL. Ni content and post-annealing temperature of nano-AFL are identified as key factors to determine the

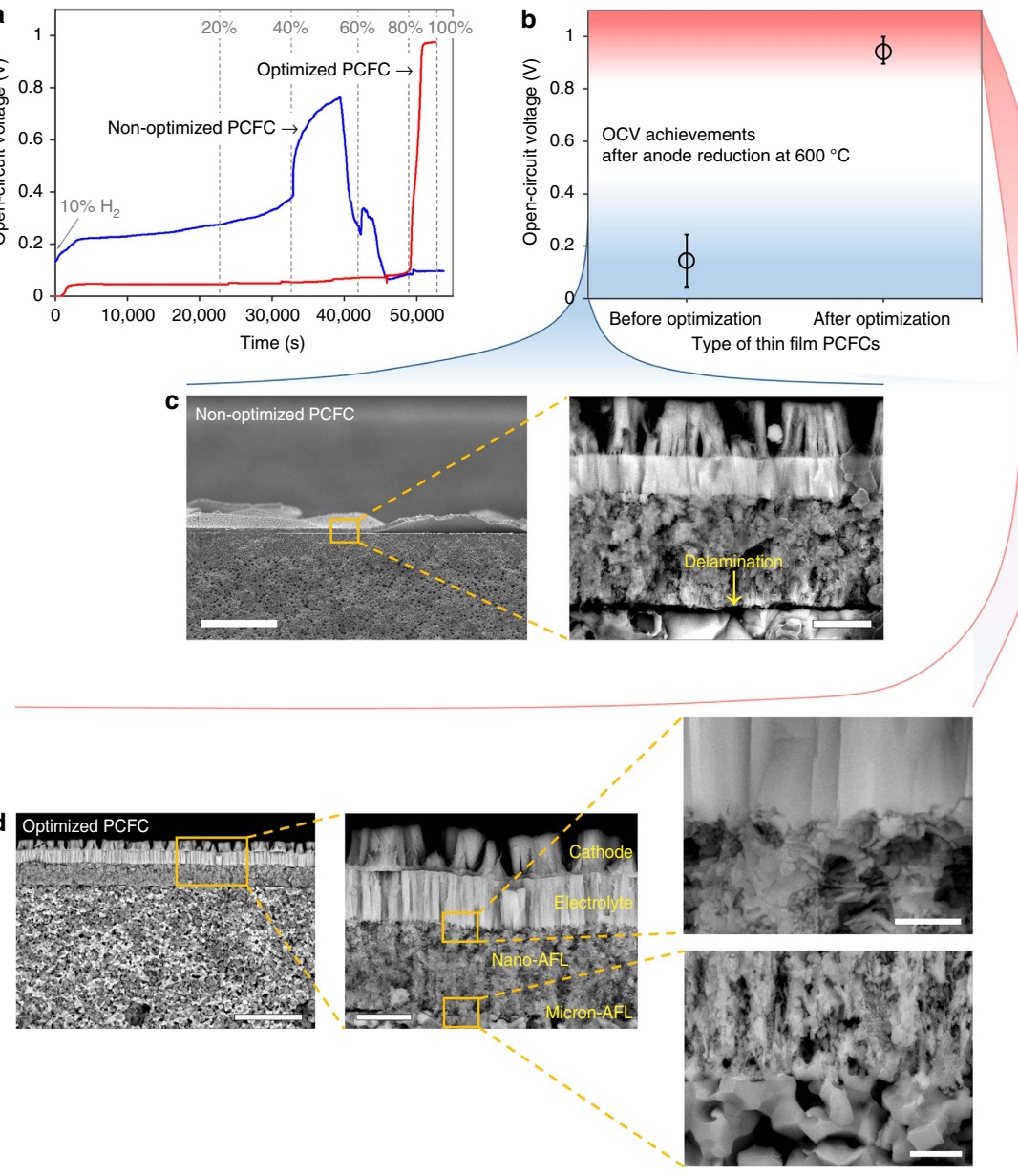

**Figure 4 | Comparison between PCFCs with optimized and non-optimized nano-AFLs.** The optimized nano-AFL was fabricated by post annealing at 1,300 °C for 1h after PLD with a volumetric composition of 50:50 (Ni:BZY), while the non-optimized nano-AFL by post annealing at 1,200 °C for 1h. (**a**) OCV profiles obtained during anode reduction in which $H_2$ concentration was varied from 0% to 100% with $N_2$ balanced in the feeding gas at the anode side. (**b**) OCV achievements after the reduction at 600 °C obtained from repeating measurement of the PCFCs fabricated under the same conditions with the PCFCs used in **a**, and error bars present the gap between the maximum and minimum values. (**c**) SEM images of the PCFC fabricated under non-optimized conditions exhibiting poor adhesion between nano- and micron-AFLs after reduction. Scale bars from left, 100 and 2 μm, respectively. (**d**) SEM images of the optimized PCFC after reduction exhibiting fully integrated morphologies. Scale bars from left and top, 10, 2, 0.5 and 1 μm, respectively.

microstructural factors. From the optimization, it was concluded that the most satisfactory quality of nano-AFL is obtained when the nano-AFL contains 50 vol% Ni and is post annealed at 1,300 °C. Detailed discussion on the optimization of the nano-AFL is in the Supplementary Materials. By applying optimized NiO–BZY nano-AFL over the micron-AFL, the surface of the anode is now covered with grains with diameter ∼ 100 nm (Fig. 3c) and the size of open pores is also much reduced in comparison with that of the micron-AFL after the anode reduction (Fig. 3d).

The impacts of the anode optimization, particularly focusing on nano-AFL, are clearly compared in Fig. 4. The open-circuit voltage (OCV) profiles in Fig. 4a were obtained from two

different PCFCs during the anode reduction with varying $H_2$ concentration from 0 to 100% in $N_2$ valance. The first PCFC adopted nano-AFL fabricated under the optimal condition (50 vol% Ni and is post annealed at 1,300 °C) and the second PCFC used a non-optimized condition, with a 100 °C lower post-annealing temperature. An irreversible OCV drop appears in the PCFC fabricated under the non-optimized conditions, whereas the OCV of the optimized PCFC sharply increased after the 80% $H_2$ reduction step. Only the optimized PCFC eventually reached high OCVs close to the theoretical value of BZY considering the transference number combined the electric and ionic transports (∼ 1.08 V at 600 °C)[11].

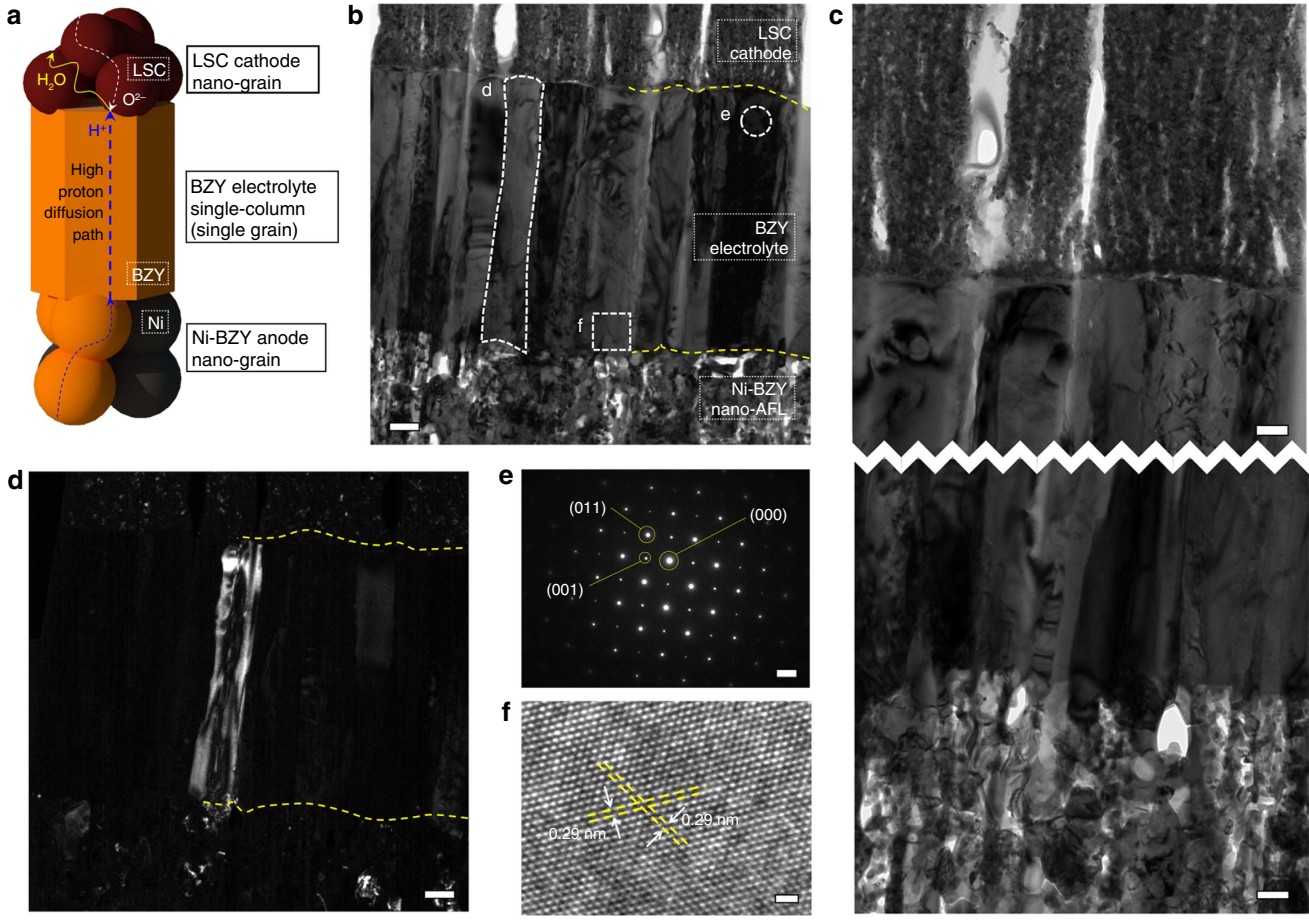

**Figure 5 | TEM characterization on the optimized PCFC.** (**a**) A schematic diagram of single column in the thin BZY electrolyte and the neighbouring electrode grains in the fuel cell configuration with possible charge transport path. (**b**) Bright-field image of dense BZY electrolyte in the middle and nano-porous electrodes. The top and bottom layers are LSC cathode and Ni–BZY nano-AFL, respectively. Scale bar, 0.2 μm. (**c**) Higher magnification of bright-field image at the interfaces between the electrolyte and the electrodes, clearly showing the grain structure of each elements. Scale bars, 0.1 μm. (**d**) Dark-field image of the area shown in **b**. The highlighted single column demonstrates it contains a single grain. Scale bar, 0.2 μm. (**e**) A SAED pattern deduced from the marked area in **b**, which matches with cubic perovskite BZY. Scale bar, 2 nm$^{-1}$. (**f**) HR-TEM image of the marked area in **b** showing the lattice images. Scale bar, 1 nm.

To check the reproducibility of the OCV values, at least three PCFCs fabricated at the identical condition were tested in the optimization process (Fig. 4 and Supplementary Fig. 6). As the result, high OCVs with small scatter were obtained from the optimal PCFCs, indicating that the thin and dense BZY electrolytes can be reproducibly fabricated on the optimized anode structure. In contrast, the PCFCs with non-optimized nano-AFLs always yielded poor OCVs. The reason of this difference between the two types of PCFCs is revealed from post-mortem SEM observation (Fig. 4c,d). The cross-sectional SEM images of the non-optimized PCFC show delamination between nano- and micron-AFLs (Fig. 4c), indicating the poor adhesion of the nano-AFL and the powder-processed anode surface. This delamination is expected to accompany local cracks through the membrane, resulting in the abrupt OCV drop with crossover of hydrogen during the reduction step shown in Fig. 4a. It appears that the annealing temperature of 1,200 °C is insufficient to develop interfacial adhesion by connecting BZY grains between nano- and micron-AFLs due to the poor sinterability of BZY. On the other hand, good interfacial adhesion was observed in the cross-section SEM images of the optimal PCFC, which would ensure both ionic and electronic paths through the entire anode (Fig. 4d).

It should be noted that high OCV was observed in the optimized cell at high concentration of hydrogen (Fig. 4a). We suspect that this is due to the structural characteristics of nano-AFLs, which comprises multiple layers with well-ordered nano-size pores, as shown in Fig. 4d. This nanoporous structure is favourable for sustaining thin and dense BZY electrolytes and for promoting the charge-transfer reaction at electrolyte–electrode boundaries. However, it is also anticipated that getting effective gas supply thoroughly through the layers could be challenging through such small pores. Therefore, opening up these small pores by reduction throughout the nano-AFLs could be retarded significantly, especially when low-concentration hydrogen is used. Moreover, the supply gas should compete against the counterflow of the water outgas that is a product of NiO reduction, which implies that hydrogen delivered near the electrolyte could be diluted more. In the case of non-optimized nano-AFLs, however, relatively large-scale cracks or spaces between the delaminated layers form, as shown in Fig. 4c, where the hydrogen supply gas could be delivered more effectively through these large spaces and thus competition against counterflow water outgas should be less severe. For this reason, OCV of the non-optimized PCFC appeared at a relatively early stage with a relatively low concentration of hydrogen, as observed in Fig. 4a.

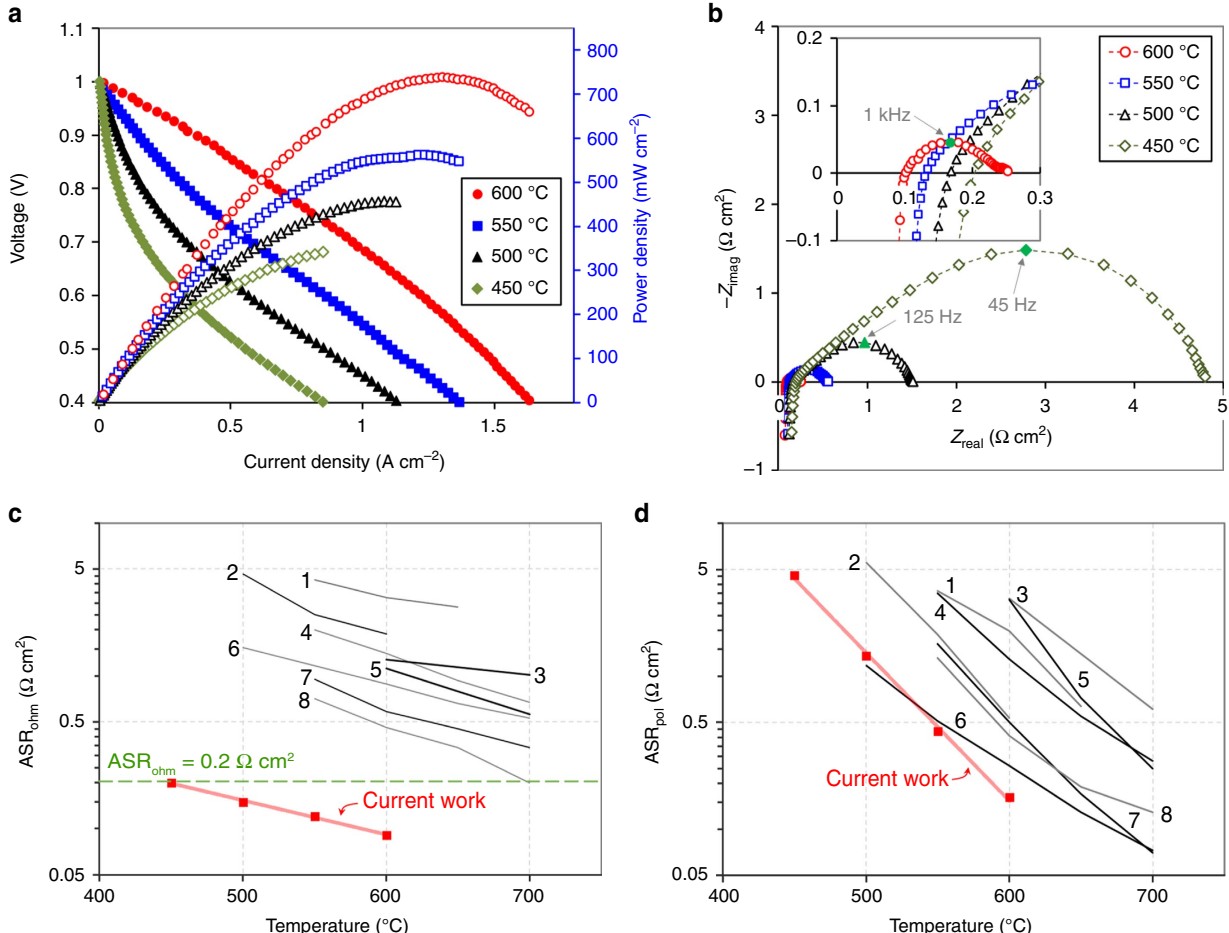

**Figure 6 | Electrochemical characteristics of the optimized PCFC.** (**a**) *I–V–P* curves obtained from an anode-supported PCFC with thin BZY electrolyte fabricated by the proposed configuration at a temperature range of 450–600 °C. (**b**) AC impedance spectra at each temperature under OCV conditions. (**c**) Ohmic area-specific resistance estimated from the impedance spectra in **b**, compared with the data of representative anode-supported BZY-PCFCs in the literature (1. Xiao *et al.*[40]; 2. Pergolesi *et al.*[41]; 3. Luisetto *et al.*[42]; 4. Sun *et al.*[43]; 5. Bi *et al.*[44]; 6. Bi *et al.*[45]; 7. Sun *et al.*[46]; 8. Sun *et al.*[47]). (**d**) Polarization area-specific resistance estimated from the impedance spectra in **b**, compared with the data of representative anode-supported BZY-PCFCs from the same studies in **c**.

**Microstructural characteristics of the optimized PCFC.** Figure 5a shows a schematic of a single columnar grain in the thin BZY electrolyte and a LSC (La$_{0.6}$Sr$_{0.4}$CoO$_{3-\delta}$) cathode and a Ni–BZY anode contacting each side of the BZY column. The schematic is drawn based on the transmission electron microscopy (TEM) analyses shown in Fig. 5b–f. First, highly dense BZY electrolyte is observed in the bright-field TEM image in Fig. 5b. In Fig. 5b, nano-porous LSC and Ni–BZY layers are also shown as top and bottom layers, respectively. The dense or porous structures of the each layer are more clearly shown in the images of a higher magnification (Fig. 5c). In the dark-field TEM image (Fig. 5d), it is confirmed that the columnar structure of the BZY electrolyte is a single grain, which does not have grain boundaries impeding the proton transfer path from the anode to the cathode. The selected area electron diffraction (SAED, obtained from the marked area in Fig. 5b) revealed that the BZY electrolyte is fully crystallized, single-phase cubic perovskite BZY (Fig. 5e). From the high-resolution-TEM (HR-TEM) image in Fig. 5f the lattice spacing of 0.29 nm can be obtained and it is in a good agreement with the (110) plane spacing of BZY[36,37]. The X-ray diffraction and SEM-energy dispersive X-ray spectroscopy (EDS) measurement of the BZY electrolyte fabricated using the same PLD conditions on sapphire substrates have confirmed that the stoichiometry matched well to that of one of the PLD targets with no secondary phase, as represented in Supplementary Fig. 8.

The high proton conduction in BZY single grain (bulk) has been identified in many studies, superior to those of the other protonic ceramics[11,14,17,38]. In recent, the exceptionally high conductivity from the epitaxial BZY thin films grown on MgO single-crystal substrates[12,13,22] raised the expectation to obtain highly performing BZY-based PCFCs by extremely limiting the numbers of the impeding grain boundaries. Until now, however, it has been extremely challenging to eliminate the grain boundaries encountering the current flow direction in the full cell, both by the powder processing and thin-film deposition. For the former, the electrolytes with very small grains and thus very high grain-boundary density are generally obtained because of the poor sinterability of BZY, and for the latter, it has been nearly impossible to deposit gas-impermeable thin BZY electrolyte over the porous anode support. Therefore, the results shown in Fig. 5 have significant importance, because these demonstrate that it is possible to realize the grain-boundary-free BZY electrolyte in the direction of proton transport by using a thin-film deposition technique and by adopting the multi-scale anode structure. Moreover, the nano-sized electrode grains are expected to improve the performance, providing sufficient electrode reaction sites on the both sides of the electrolyte.

**Electrochemical characteristics of the optimized PCFC.** The electrochemical performances of the BZY PCFC fabricated under

the optimal conditions are depicted in Fig. 6a–d. In Fig. 6a, a drop in the voltage at a low current is observed at $<500\,°C$, whereas a fall curve at a higher current is observed at $600\,°C$. This is because the electrode response is limited to other factors at different temperatures. Specifically, charge transfer reactions are considered to dominate overall electrode kinetics at low temperatures. A temperature increase to $600\,°C$ is expected to help improve the rate of electrochemical reactions and mass diffusion can dominate the electrode process because the reactants can still undergo transfer through small pores present in the nano-AFL. The power output reached a maximum of $740\,mW\,cm^{-2}$ at $600\,°C$ along with values of 563, 457 and $342\,mW\,cm^{-2}$ at the other temperatures of 550, 500 and $450\,°C$ (Fig. 6a). This power achievement is enhanced significantly compared with data from previously studied BZY-based cells, confirmed in Fig. 1 and supplementary Table 1, and greater than record data from all PCFCs previously developed ($650\,mW\,cm^{-2}$ at $600\,°C$)[39]. The OCV values were about 1.0 V, which can be considered to be in a reasonable range compared to that of the previously reported BZY-based PCFC[40–47], especially considering the low thickness of the electrolyte. It implies that the thin BZY electrolyte has the appropriate structural integrity to function as an electrolyte. However, the OCV is rather insensitive to temperature change, which may originate from certain leakage issues such as sealing. The performance improvement attributes to the results of the well-designed fuel cell configuration and its optimization as previously discussed above.

Figure 6b presents AC impedance spectra obtained at each temperature under OCV condition. Due to the complexity and many processes involved in the whole fuel cell reactions, subdivided interpretation is difficult from the impedance data, but ohmic and polarization resistances were able to be estimated. The intersection points with $x$ axis at the high- and low-frequency regime were used for the ohmic and polarization area-specific resistances (ASRs), respectively.

To examine the significant improvement of electrochemical performance, the ohmic and polarization ASRs of representative BZY-PCFCs found in the literature were compared (Fig. 6c,d). An order of magnitude lower ohmic ASRs were achieved in the current work compared to the reference values, as shown in Fig. 6c. These results suggest that the significantly reduced thickness of the BZY electrolyte is the main cause of the improved cell performance. The improvement in bonding between the porous anode and the thin and dense columnar BZY layer, as shown in Fig. 5, also seems to have contributed to the reduction in ohmic ASRs. Relatively low-polarization ASRs were also observed during the comparison (Fig. 6d). We believe that the nano-size grains of the LSC cathode and the Ni-BZY nano-AFL increased the number of active sites in the electrode reaction. Further improvement is expected by use of a high-performing and stable cathode material substituting for the LSC that has negligible proton conductivity[48]. Moreover, the improved integration of electrolyte and anode support by adoption of the multilayered AFLs using multistep post-annealing has been observed clearly in the cross section of the stack, as presented in Figs 4d and 5, which is considered to have contributed significantly to the improved charge-transfer reaction, decreased polarization ASRs and enhanced fuel cell power.

## Discussion

To fabricate highly efficient and physically/chemically stable PCFCs, an anode-supported fuel cell configuration based on BZY thin films is demonstrated in the current study. The multi-scale anode structure with reducing grain and pore sizes is confirmed to provide flat surface favourable to thin-film deposition as well as improve physical integration. On the anodes, a grain-boundary-free columnar BZY electrolyte with significantly reduced thickness was successfully fabricated by PLD. This thin BZY electrolyte is believed to substantially reduce the ohmic resistance compared with those of BZY-PCFCs quoted in literature, which is the main reason for the cell performance enhancement. The nano-porous electrodes clearly shown by TEM images were also sufficient to implement low-polarization resistance, providing increasing reaction sites on the both side of the electrolyte. As results, significantly improved power outputs were obtained from the fuel cell configuration with the maximum power density of $740\,mW\,cm^{-2}$ at $600\,°C$ that has not achieved from the other BZY-based PCFCs so far. This performance improvement using BZY provides an opportunity for practical use of PCFCs potentially solving the conflicting challenges between high performance and chemical stability that have been faced in PCFCs until now.

## Methods

**Preparation of PCFCs with thin-film BZY electrolytes.** Tape-casted NiO–BZY composites (a Ni:BZY volume ratio of 40:60 in the solid content after reduction; composition of the anode BZY powder: $BaZr_{0.85}Y_{0.15}O_{3-\delta}$) were sintered at $1{,}450\,°C$ for 10 h in air and used as the anode support. Micron-AFL tape sheet (10 μm in thickness) was placed on the porous anode body tapes containing 30 vol% polymethyl methacrylate pore formers and laminated with a cell size of $1\times1\,cm^2$. After the sintering of the anode support, surface grinding was treated to remove the NiO particles protruded from the sintered surface. Then, nano-AFLs ($\sim3\,μm$ in thickness) were deposited by PLD with a 50 vol% Ni containing NiO–BZY target. A KrF excimer laser ($\lambda=248\,nm$, Compex Pro 201 F, Coherent) was used as the ablation source with a laser fluence of $\sim2.5\,J\,cm^{-2}$ and a repetition rate of 10 Hz. The substrate temperature, $O_2$ background pressure, and target-to-substrate distance were kept at $750\,°C$, 6.67 Pa, and 5 cm, respectively, during the deposition. The nano-AFLs were post annealed in ambient air at $1{,}300\,°C$ for 1 h with a uniform heating and cooling rate of $2\,°C\,min^{-1}$.

Dense BZY electrolyte layers (2.5 μm in thickness) were deposited under the same PLD conditions used for nano-AFLs. Validity of this process for growing BZY films is discussed rigorously and confirmed in our previous work[49]. The deposited BZY electrolytes were followed by annealing at $1{,}200\,°C$ for 3 h to improve adhesion at the interface with the anode support. Porous LSC (2 μm in thickness) was deposited as the cathode by PLD at room temperature with an $O_2$ pressure of 13.3 Pa and an area of $0.3\times0.3\,cm^2$. This process was followed by annealing at $650\,°C$ for 1 h to form a porous morphology.

**Fuel cell test.** Before operating the fuel cell, reduction of the anode was performed by gradually increasing the $H_2$ concentration from 0 to 100% with $N_2$ as the balance gas at $600\,°C$ for 9 h while measuring the OCVs every 10 s. Humidified $H_2$ gas (3% $H_2O$) was flowed on the anode side at $50\,ml\,min^{-1}$, and air was fed as the oxidant on the cathode side at the same flow rate during the test. An Au mesh and Ni foam were placed on the cathode and anode surfaces, respectively, for current collection, and a commercial alumina paste (P-24, Toku Ceramic) was used for gas sealing. The $I$–$V$ and AC impedance data were collected at $450$–$600\,°C$ using the Gamry framework system (Gamry Reference 3000 Potentiostat/Galvanostat/ZRA). The impedance data were obtained in the frequency range of $10^6$–$0.1$ Hz with an amplitude of 10 mV under OCV condition. The data were analysed using Z-view software (v3.4c, Scribner Associate Inc.).

**Microstructure observation.** The prepared anode supports or NiO–BZY nano-AFLs deposited on them were placed in a tube furnace under the flow of 4% $H_2$–Ar at $650\,°C$ for 10 h to investigate the morphology changes of nano-AFLs resulting from reduction. SEM (XL-30 FEG, FEI) was utilized to observe morphologies of the anode surface and the full cell surface and cross-section. To investigate in-depth microstructure crystallinity of the thin BZY electrolyte and its near anode and cathode grains, TEM (Tecnai F20, FEI) was used. Focused ion beam (Helios NanoLab 600, FEI) was used to prepare the TEM sample.

**Data availability.** The authors declare that the main data supporting the findings of this study are available within the article and its Supplementary Information files. Extra data are available from the corresponding author upon request.

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

## Acknowledgements

This work was supported by a National Research Foundation of Korea (NRF) Grant funded by the Korean Government (Grant No. NRF-2013R1A1A1A05013794, 2016R1D1A1B03932377) and the Brain Korea 21 Plus program (Grant No. 21A20131712520). We are also grateful to the Global Frontier R&D Program at the Center for Multiscale Energy Systems (Grant No. NRF-2015M3A6A7065442) of the National Research Foundation (NRF) of Korea funded by the Ministry of Science, ICT & Future Planning (MSIP) and to the Institutional Research Program (2E26081) of Korea Institute of Science and Technology (KIST) for financial support.

## Author contributions

K.B., J.-W.S. and J.H.S. planned this study and co-wrote the manuscript. K.B. carried out the experiments and the characterizations. D.Y.J., H.J.K. and D.K. conducted the electrochemical measurements. J.H. and B.-K.K. advised in the interpretation of data regarding the physical properties. J.-H.L. advised in the interpretation of data regarding the electrochemical properties. All authors read and commented on the manuscript.
