## [Peer Review File · Nature Communications]

Reviewers' comments:

Reviewer #1 (Remarks to the Author):

This paper describes the use of Y-doped BaZrO₃ (BZY), an excellent chemical stability electrolyte for low temperature SOFCs. The preparation of columnar structure allows the BZY electrolyte shows high conductivity. As a result, the BZY cell in this study shows a record high performance with the maximum power density of 740 mW cm⁻² at 600 oC. The highly-performance improvement using BZY seem to give a great opportunity to use of PCFCs working at LTs. However, in my opinion, this manuscript contains some scientific flaws which should be solved before it can be accepted for publication.

1. Figure 6(a). In the SOFC system, the OCV values normally decrease with increasing temperatures. However, the OCV values in the study almost keep constant and it is not influenced by the temperatures. This is very strange.

2. Figure 6 (a) and (b). From the I-V curves, we can find the cell performance is high. However, if we look at the impedance shown in Figure 6 (b), we can find the strange results. For example, the resistance of the cell at 450 oC is almost 20 times larger than that at 600 oC, but the power density at 450 oC is only reduced to half compared with that at 600 oC. Is it normally? Actually, we can estimate the power density from the impedance of the cell and OCV values. The estimated peak power density at 450 oC is only about 50 mW cm⁻², not as high as the reported value of 342. It is the same for the values reported at 500 and 550 oC. In addition, the total ASR is 0.5 Ω cm² (=ASR_{ohm} (0.09 Ω cm²) + ASR_{pol} (0.41 Ω cm²) at 600 oC in Supplementary Table 1 which is not in accordance with the value shown in Figure 6(b) (seen from figure. 6(b), the total ASR is about 0.25 Ω cm² at 600 oC). The composition of BZY is BaZr_{0.85}Y_{0.15}O_{3-δ} (line 109) in the manuscript which is not identical with BaZr_{0.9}Y_{0.1}O_{3-δ} in Supplementary Table 1. It is puzzling. I suggest the authors check the raw testing data.

3. Figure 6(c). The authors show their ohmic ASR is much smaller than that in literatures and want to demonstrate their columnar structure BZY electrolyte having high conductivity. I have to say the authors are less rigorous. They do not mention the electrolyte thickness in their comparison and the electrolyte thickness is critical for the ohmic ASR. The BZY electrolyte thickness in this study is only about 2 micrometers, which is much smaller than that in the literatures listed by the authors. Figure 6(c) is meaningless if they are not put in the same thickness. In fact, some literatures data show advantages over this study if they are put at the same thickness.

4. Figure 6(d). This time the authors know to put the comparison of ASR under the same electrolyte thickness. It is good, but authors select the literature reports benefiting them. The BZY electrolyte they selected shows low conductivity. In reality, the BZY electrolyte conductivity has been improved dramatically in the past several years. The studies at Ref. 14, 36, 41 and Ref. 23 in the supporting information provide different strategies to prepare highly conductive BaZrO₃ electrolyte with simple ceramic processing method. If the authors put these literatures in the figure, we can see these BZY electrolytes show smaller ohmic ASR compared with the current study. The strategy proposed by the authors does not have any advantages compared with the traditional methods. In another word, the BZY film in this study is not very highly conductive as claimed by the authors.

There are another two questions:

1. Authors use LSC as the cathode leading to TPBs are only located at the electrolyte/cathode interface, which is not an ideal structure for proton-conducting SOFCs. So, my question is: the cathode material is not the best (I would like to say it is a poor choice for proton-conducting SOFCs), the BZY electrolyte conductivity is not the best. How the authors get the record high performance? Is it just because of the use of very thin electrolyte (around 2 micrometer)?

2. This method of preparation of PCFCs with thin-film BZY electrolytes need several sintering and post-annealing processes at high temperature, I think it is hard to use for practical application of PCFCs working at LTs.

Reviewer #2 (Remarks to the Author):

report on paper by K Bae et al:

This paper reports on the preparation of proton-conducting Y-doped BaZrO₃ as electrolyte for intermediate-temperature fuel cells as a dense 2-3 micron thick film with columnar structure (on an initially porous substrate as it has to be used in a fuel cell!). This avoids the detrimental effect of blocking grain boundaries, and thus allows to demonstrate the full potential of proton-conducting Y-doped BaZrO₃ as fuel cell electrolyte. This paper reports increase performance over previous publications (e.g. [36]), but more importantly it is the first study that demonstrates a fuel cell based on proton conducting oxides where the electrolyte resistance is smaller than the electrode polarization resistance. I.e. by then optimizing the electrode materials/microstructures a further increase of performance is expected. Personally, I would rather emphasize this aspect in the title (e.g. "demonstrating the potential of Y-doped BaZrO₃ for fuel cells") than the present title. Since blocking grain boundaries are an issue also in other materials, this paper is clearly relevant beyond the community of proton-conducting oxide fuel cells.

Nevertheless, some points should be improved before publication:

- pg 3 line 49: when discussing the blocking gb in proton-conducting perovskites (or the high bulk proton conductivity), earlier papers should be cited, e.g. Kreuer SSI 125 (1999) 285
- pg 4 line 56: for the use of ZnO sintering aid, also Irvine Adv Mat 18 (2006) 3493 should be cited
- pg 4 line 56: here also the "solid state reactive sintering" approach by R O'Hayre should be mentioned
- ref. [36] with the max. power reported before the present study should be mentioned already in the context of fig. 1. It is a Ce-containing electrolyte (not pure Y-doped BaZrO₃) but it also represents a milestone on the way to find a processing method largely solving the issue of the blocking gb. It would also be good to include the data for the best cells from [36] into fig 6 for comparison
- fig 4: why does the OCV of the optimized cell increase only at high H₂ concentrations, in contrast to the other curve? please explain
- pg 9 line 167: it is clear that electrode delamination decreases performance, but it needs more explanation why it should decrease OCV
- the optimization of the anode and anode functional layer is discussed in great detail (it is an important part as it is the key to fabricating a thin gastight electrolyte on a porous anode substrate). However, also some more details about the PLD process for the electrolyte deposition would be welcome (e.g. how sensitive it is to variations in PLD processing conditions)
- pg 14 line 255 "non proton conducting LSC": I think for LSC there are no data yet, but for other materials there are indications that typical cathode perovskites have a small but nonzero proton solubility and corresponding proton conductivity that expands the reactive zone beyond the triple phase boundary. see e.g. Han (SSI 181 (2010) 1601) and Poetzsch (PCCP 16 (2014) 16446)
- experimental: please (roughly) indicate the grain sizes of the starting materials. what is the Y-content in the anode layer? what is the lateral cell size? full details for PLD are required (pulse length, repetition frequency, target substrate distance...)

Reviewer #3 (Remarks to the Author):

For a paper to be published in a venue as prestigious as Nature Communications it should encompass either a remarkable technological breakthrough, or an exceptional scientific understanding, or ideally, both. This paper sells itself as one in which a technological breakthrough has been achieved on the basis of a scientific understanding and rational solution to an obstacle.

The basic premise of this work is that the high grain boundary impedance has been the cause of poor performance in BZY electrolyte fuel cells, and that the columnar grains of PLD grown BZY films circumvent this problem, leading to exceptional performance. Both the premise (that the grain boundaries have been the culprit) and the final conclusion (that the performance is

exceptional) are disputable.

The fuel cell performance reported here by Bae et al. is quite impressive. However, in light of the recent work of Duan et al. (reference 36), it is perhaps less impressive than the authors make claim. As recognized by the present authors, Duan et al. achieved at 600C a peak power density of 650 mW/cm² as compared to the present result of 740 mW/cm². An increase by about 15% over what was a breakthrough result deserves recognition, but within the appropriate context. In Figure 1, in which the context is presented, the authors apparently limit the cells represented to those with an electrolyte of at least 70% Zr on the B site. This limitation makes sense if one wants to try and understand why barium zirconate based cells have hit a performance limitation, but it does not make sense if one is trying to claim a technological breakthrough. If an electrolyte, such as Ba(Ce_{0.7}Zr_{0.1}Y_{0.1}B_{0.1})O₃ + 1%NiO as employed by Duan et al., provides attractive performance and greater ease of processability, then the present authors need to explain how their work represents a technological advantage. One point is that BCZYYb is not stable when methane is supplied as the fuel. The present authors have not capitalized on this (likely) advantage of BZY, stability in the presence of carbon-containing fuels. Nor have they explored whether a Ni-BZY containing anode would be immune to carbon deposition when so operated.

The conclusion that grain boundary impedance has limited power output from barium zirconate based fuel cells is based on (1) the observation that no-one has been able to make a good cell from these electrolytes (Figure 1, Table S1), and (2) that the ohmic resistance in cells from the literature which have been characterized by impedance spectroscopy is greater than what one would expect based on the bulk (grain interior) conductivity of the electrolyte material and its thickness, Figure 6(c). This evidence is weak. First, any number of reasons could lead to poor cell performance, so Table S1 alone does not provide evidence of the negative role of grain boundaries. Figure 6(c) is somewhat more convincing, but still weak, as an offset in an impedance spectrum could be due to factors other than grain boundary impedance. As shown by Yamazaki et al. (ref 14) for a typical BZY material with grain size on the order of 0.5 μm, the grain boundary impedance will be about 50% of the total impedance at 600C. Therefore, eliminating grain boundaries as done in the present work would effectively double the macroscopic conductivity. This is not enough to explain the poor performance of literature cells, or the high ohmic offsets previously reported. It is also important to recognize that the literature data shown in Figure 6(c) originate from just two laboratories, so any systematic error in those measurements would be exaggerated as a general literature result. Furthermore, because some of the results summarized in Table S1 also reflect PLD grown electrolytes and such growth almost always produced columnar films with few grain boundaries normal to current flow, it is difficult to imagine that those cells suffered from grain boundary impedance. The authors are encouraged to consider other factors that may have led to the high performance observed here. For example, is the stoichiometric control of the electrolyte particularly good? It is known that Ba deficiency can result in low conductivity. Or are the electrode-electrolyte contacts particularly good? It is plausible that PLD growth of the various layers provides better contacts than co-sintering of heterogeneous components.

While I am critical of this work in terms of the interpretation of why the performance is excellent, and I am concerned that there is a bit of exaggeration of the extent to which this is truly a breakthrough, the work is nonetheless impressive and worthy of recognition. I do not believe it is absolutely essential that the authors understand definitively why others have been rather successful in pursuing BZY-based cells in order to proceed to publication. However, a revision must acknowledge that several factors may explain the high performance of their cells relative to others. In addition, the work of Duan et al must be discussed early in the manuscript in terms of context, and the authors need to make a convincing case as to why the BZY cells with complex PLD processing steps would be technologically preferable to BCZYYb. I don't believe a 15% performance gain alone would warrant the processing effort.

Reviewers' comments:

Reviewer #1 (Remarks to the Author):

This paper describes the use of Y-doped BaZrO₃ (BZY), an excellent chemical stability electrolyte for low temperature SOFCs. The preparation of columnar structure allows the BZY electrolyte shows high conductivity. As a result, the BZY cell in this study shows a record high performance with the maximum power density of 740 mW cm⁻² at 600 °C. The highly-performance improvement using BZY seem to give a great opportunity to use of PCFCs working at LTs. However, in my opinion, this manuscript contains some scientific flaws which should be solved before it can be accepted for publication.

1. Figure 6(a). In the SOFC system, the OCV values normally decrease with increasing temperatures. However, the OCV values in the study almost keep constant and it is not influenced by the temperatures. This is very strange.

: An OCV close to 1 V generally represents good separation of electrodes, minor transfer of electrons/holes across the layer, and gas-tightness. In this respect, we consider our membrane to be free of problematic flaws. Compared to the reference data from the literature (supplementary Table 1), our OCV falls in a reasonably decent range. According to our calculation, OCV is supposed to decrease by 0.02 V if the temperature changes from 450 to 600 °C at the given conditions. This small change is often missed in real measurements as the perfect “zero” current state where the cell voltage represents OCV is hard to achieve in practice. We agree that the OCV-temperature trend is not observed clearly in our experiment. However, we believe that reasonably high OCVs sufficiently support the reliability of our experimental setups, quality of our samples, and credibility of our *I-V* data. This point was added to the revised manuscript as below:

- line 265 on page 15,

“For all the measured temperatures, high OCVs (over 1 V) were observed, implying proper electronic insulation and gas-tightness of the thin BZY electrolyte.”

2. Figure 6 (a) and (b). From the I-V curves, we can find the cell performance is high. However, if we look at the impedance shown in Figure 6 (b), we can find the strange results. For example, the resistance of the cell at 450 °C is almost 20 times larger than that at 600 °C, but the power density at 450 °C is only reduced to half compared with that at 600 °C. Is it normally? Actually, we can estimate the power density from the impedance of the cell and OCV values. The estimated peak power density at 450 °C is only about 50 mW cm⁻², not as high as the reported value of 342. It is the same for the values reported at 500 and 550 °C. In addition, the total ASR is 0.5 Ω cm² (=ASR_{ohm} (0.09 Ω cm²) + ASR_{pol} (0.41 Ω cm²)) at 600 °C in Supplementary Table 1 which is not

in accordance with the value shown in Figure 6(b) (seen from figure. 6(b), the total ASR is about $0.25 \Omega \text{ cm}^2$ at $600 \text{ }^\circ\text{C}$). The composition of BZY is $\text{BaZr}_{0.85}\text{Y}_{0.15}\text{O}_{3-\delta}$ (line 109) in the manuscript which is not identical with $\text{BaZr}_{0.9}\text{Y}_{0.1}\text{O}_{3-\delta}$ in Supplementary Table 1. It is puzzling. I suggest the authors check the raw testing data.

: It is not clear how the peak power density of 50 mW cm^{-2} is estimated from OCV and ASRs at OCV. The power density from the I-V can be estimated as $V*J = (\text{OCV}-\text{OP}_{\text{act}}-\text{OP}_{\text{ohm}})*J = \text{OCV}*J - (\text{ASR}_{\text{pol}}-\text{ASR}_{\text{ohm}})*J^2$, if loss from the mass transport is ignored where V represents the cell voltage, J the current density, OP_{act} the activation overpotential, OP_{ohm} the ohmic overpotential, ASR_{pol} the resistance from polarization/charge-transfer processes, and ASR_{ohm} the resistance from charge transport. In this equation, ASR_{pol} is a function of J, as the impedances at the electrode surface and the electrode-electrolyte interfaces are influenced heavily by the local concentration of charged species during the charge-transfer processes. Therefore, in our understanding, the impedance values measured at OCV provided in this work cannot be directly used for estimation of peak power density that is generally marked at relatively high current where the $\text{ASR}_{\text{pol}} = \text{ASR}_{\text{pol}}(J)$ is supposed to be reduced significantly with an increase exponentially in current (J). For the same reason, it is difficult to link the reduction of impedances at OCVs directly to the estimation of power enhancement scale.

The polarization ASR and the composition data in supplementary Table 1 were miswritten, thus those values have been corrected in the revised supplementary Table 1 as $0.16 \Omega \text{ cm}^2$ and $\text{BaZr}_{0.85}\text{Y}_{0.15}\text{O}_{3-\delta}$, respectively.

3. Figure 6(c). The authors show their ohmic ASR is much smaller than that in literatures and want to demonstrate their columnar structure BZY electrolyte having high conductivity. I have to say the authors are less rigorous. They do not mention the electrolyte thickness in their comparison and the electrolyte thickness is critical for the ohmic ASR. The BZY electrolyte thickness in this study is only about 2 micrometers, which is much smaller than that in the literatures listed by the authors. Figure 6(c) is meaningless if they are not put in the same thickness. In fact, some literatures data show advantages over this study if they are put at the same thickness.

: The reason why we compared ASRs in Fig. 6c, and not conductivity, is that our major achievements include not only reduction of grain boundaries along the membrane thickness (reduction of total conductivity) but also successful fabrication of thin and dense BZY layers well-integrated on a porous anode support. As reported in many studies, sintering of dense BZY is very challenging. In this work, we have demonstrated successful fabrication of thin and dense BZY membranes as working electrolyte for anode-support type PCFCs and confirmed that PCFCs with this thin BZY electrolyte can be truly useful, confirming high performance in intermediate temperature ranges. It should be noted that many researchers have been skeptical about using BZY for fuel cell electrolyte, as all of the studies before ours have failed to produce high power with BZY fuel cells, to our knowledge [Refs. 15, 42, 43, 44, 45], although BZY has demonstrated high bulk conductivity and excellent chemical stability, especially against moisture and carbon compounds in material characterization

[Refs. 6, 9]. More detailed discussion about this aspect and the consequent revision to address it are continued in the response to the next comment.

4. Figure 6(d). This time the authors know to put the comparison of ASR under the same electrolyte thickness. It is good, but authors select the literature reports benefiting them. The BZY electrolyte they selected shows low conductivity. In reality, the BZY electrolyte conductivity has been improved dramatically in the past several years. The studies at Ref. 14, 36, 41 and Ref. 23 in the supporting information provide different strategies to prepare highly conductive BaZrO₃ electrolyte with simple ceramic processing method. If the authors put these literatures in the figure, we can see these BZY electrolytes show smaller ohmic ASR compared with the current study. The strategy proposed by the authors does not have any advantages compared with the traditional methods. In another word, the BZY film in this study is not very highly conductive as claimed by the authors.

: As mentioned in the response to the reviewer's comment 3 above, our major achievements include reduction of grain boundary and grain boundary impedances by adoption of "single-grain-columnar structure" and fabrication of thin and dense BZY electrolyte that is well-integrated on porous anode support. As a result, the highest power output ever reported in PCFCs has been accomplished, confirming promising potentials of the BZY material as an "effectively working" electrolyte for high performance PCFCs for the first time. We agree that ohmic ASR can be reduced further, considering the high conductivity of bulk BZY or BZY in large grains reported in the literature [Refs. 12, 13, 14, 20, 21, 22]. However, this expectation can be realized only when BZY is fabricated as an effectively thin membrane on a practically usable form of fuel cells. We doubt that the strategies used by the quoted studies, such as nano-size-powder processing, long-time sintering, or deposition on single-crystal substrates, are truly capable of fabricating an effectively thin electrolyte membrane in a working fuel cell and able to demonstrate good fuel cell performance. We agree that our fabrication method is not simple or advantageous over other "traditional" powder or slurry-based processes. Our focus is, however, on confirming the effectiveness of the BZY material as a practically "working" fuel cell electrolyte. Our conclusion is that the effectively thin and dense membrane in the "single-grain-columnar structure" is key for high performance of BZY-PCFCs. Therefore, if a simple method for production of our anode-support PCFC with thin film BZY can be developed, it would be great follow-up to this work. We acknowledge and accept that the methods we use in this work are not attractive for industrial-level production; however, that is not the focus of our main claims or accomplishments.

The figure below represents the "calculated" conductivity of our BZY membranes, estimated from the Ohmic ASRs (Fig. 6c) and thickness of the electrolytes (thickness/ASR_{ohm}). It is compared to i) the conductivities of BZY reported or calculated using reported data from studies on "working" BZY-PCFCs and ii) "bulk" conductivity of BZY measured or estimated from studies on BZY material characterization, not from fuel cell measurement setups. Although most grain boundary impedances are effectively eliminated as observed from the TEM images, our "calculated" conductivity did not reach the bulk BZY value of such. Considering real fuel cell measurement setup, there are "inevitable" resistances other than the membrane resistance contributing to the

ohmic impedances including resistances at the electrolyte/electrode interfaces, resistances of the electrodes, resistances of the current-collecting components and devices, etc. Moreover, the relative portion of these parasitic ohmic resistances in the overall measured ohmic impedances are expected to increase with a decrease in electrolyte thickness, *i.e.*, electrolyte resistance (in other words, the “calculated” conductivity would be close to the real conductivity value of the membrane if the membrane is fairly thick.). For this reason, we did not consider this “calculated” conductivity as the “real” conductivity of our BZY membrane, and we do not use the terms ASR_{membrane} or $ASR_{\text{electrolyte}}$ but instead use ASR_{ohm} throughout the article. If there is a report about “fuel cell” performance with a BZY membrane of the same or similar thickness to ours, it would be a really good reference for comparison. To our knowledge, however, no such paper has been published yet.

From the reviewer’s comment, we realized that the readers may be confused about our main accomplishments. In the revision, therefore, we emphasized our major achievements clearly throughout the article regarding the successful fabrication of a thin and dense membrane that is well-integrated on a porous anode support and confirmation of the effectiveness of the BZY material as an electrolyte for high performance PCFCs in a fuel cell for the first time. In addition, we highlight more clearly that both reduction of thickness and elimination of grain boundary across the membrane by the “single-grain-columnar BZY structure” have contributed significantly to the relatively small ASR_{ohm} and enhancement of fuel cell power as indicated below:

- line 276 on page 15,

“To examine the significant improvement of electrochemical performance, the ohmic and polarization ASRs of representative BZY-PCFCs found in the literature were compared (Fig. 6c,d). An order of magnitude lower ohmic ASRs were achieved in the current work compared to the reference values, as shown in Fig. 6c. This finding can be attributed to the significantly reduced thickness of the BZY electrolyte and, more importantly, the achievement of an impeding-grain-boundary-free columnar structure of such. ”

There are another two questions:

1. Authors use LSC as the cathode leading to TPBs are only located at the electrolyte/cathode interface, which is not an ideal structure for proton-conducting SOFCs. So, my question is: the cathode material is not the best (I would like to say it is a poor choice for proton-conducting SOFCs), the BZY electrolyte conductivity is not the best. How the authors get the record high performance? Is it just because of the use of very thin electrolyte (around 2 micrometer)?

: We agree that LSC is not the best cathode material for evaluation of PCFCs as there is little proton conduction through this material. Although a few studies have addressed novel cathodes for PCFCs, these materials are, as of yet, not considered promising due to a lack of well-established property data supported by theoretical and experimental proof. For this reason, we decided to use LSC because it is accepted as a “standard” ceramic cathode, popularly used for evaluation of ceramic fuel cells. As claimed above, adoption of a thin and dense BZY membrane in the “single-grain-columnar structure” and good integration of this membrane on the porous anode support are considered key for achieving high fuel cell performance, demonstrating dramatically reduced ohmic ASR compared to that of other fuel cells with BZY membranes. Another positive message implied in our paper is that the high power was achieved even with this suboptimal cathode. In future work, we look forward to adopting “proven” highly-performing PC-cathodes in our cell and expect more enhanced fuel cell performance. It would be a great follow-up to this work. This point is highlighted in the revised manuscript per below:

- line 284 on page 15,

“Further improvement is expected by use of a high-performing and stable cathode material substituting to LSC that has negligible proton conductivity⁴⁰.”

2. This method of preparation of PCFCs with thin-film BZY electrolytes need several sintering and post-annealing processes at high temperature, I think it is hard to use for practical application of PCFCs working at LTs.

: As mentioned above in the response to the reviewer’s comment 4, we acknowledge that our fabrication method

is not practically advantageous. Our focus was on confirming the effectiveness of BZY material as a practically “working” fuel cell electrolyte. Moreover, as mentioned above in our response to the reviewer’s comment 3, it should be noted that many researchers have been skeptical about using BZY as a fuel cell electrolyte because of the disappointing performance from BZY-based fuel cells [Refs.6, 7, 15, 44], regardless of their high bulk conductivity and excellent chemical stability especially against moisture and carbon compounds identified in the material characterization of BZY [Refs. 6, 9]. The method used in this work is for prototyping of our proposed structure, specifically an anode-supported cell tightly integrated with an effectively thin and dense membrane in the single-grain-columnar structure, rather than for providing a good solution for practical production. Therefore, our paper should be considered a proof-of-concept report, and thus a good fit for publishing in scientific journals.

Reviewer #2 (Remarks to the Author):

This paper reports on the preparation of proton-conducting Y-doped BaZrO₃ as electrolyte for intermediate-temperature fuel cells as a dense 2-3 micron thick film with columnar structure (on an initially porous substrate as it has to be used in a fuel cell!). This avoids the detrimental effect of blocking grain boundaries, and thus allows to demonstrate the full potential of proton-conducting Y-doped BaZrO₃ as fuel cell electrolyte. This paper reports increase performance over previous publications (e.g. [36]), but more importantly it is the first study that demonstrates a fuel cell based on proton conducting oxides where the electrolyte resistance is smaller than the electrode polarization resistance. I.e. by then optimizing the electrode materials/microstructures a further increase of performance is expected. Personally, I would rather emphasize this aspect in the title (e.g. "demonstrating the potential of Y-doped BaZrO₃ for fuel cells") than the present title. Since blocking grain boundaries are an issue also in other materials, this paper is clearly relevant beyond the community of proton-conducting oxide fuel cells.

: We appreciate the suggestion regarding the title. The revised manuscript is titled as below:

- Title,

“Demonstrating the potential of yttrium-doped barium zirconate electrolyte for high performance fuel cells”

Nevertheless, some points should be improved before publication:

- pg 3 line 49: when discussing the blocking gb in proton-conducting perovskites (or the high bulk proton conductivity), earlier papers should be cited, e.g. Kreuer SSI 125 (1999) 285

: The suggested literature is included as a reference [Ref. 16] in the revision.

- pg 4 line 56: for the use of ZnO sintering aid, also Irvine Adv Mat 18 (2006) 3493 should be cited

: The suggested literature is included as a reference [Ref. 20] in the revision.

- pg 4 line 56: here also the "solid state reactive sintering" approach by R O'Hayre should be mentioned

: The suggested content is included in the revised manuscript as an alternative way to enhance grain growth of BZY as follows:

- line 58 on page 4,

“Solid-state reactive sintering, where material synthesis and sintering are carried out simultaneously using nano-size precursors, has enabled the growth of relatively large BZY grains and effectively reduced grain-boundary resistance²¹, for which the use of sintering aids (*e.g.*, NiO) that inevitably degrade chemical and electrical properties was still required.”

- ref. [36] with the max. power reported before the present study should be mentioned already in the context of fig. 1. It is a Ce-containing electrolyte (not pure Y-doped BaZrO₃) but it also represents a milestone on the way to find a processing method largely solving the issue of the blocking gb. It would also be good to include the data for the best cells from [36] into fig 6 for comparison.

: As suggested, the peak power density data from [Ref. 25] is now included in Fig. 1, and the figure caption was revised as below.

- line 80 on page 5 (figure caption of Fig. 1),

“...with the record data previously reported from a PCFC with BaCe_{0.7}Zr_{0.1}Y_{0.1}Yb_{0.1}O_{3-δ} (BCZYYb) electrolyte²⁵.”

However, information of ASR or data of estimated ASR of the highest performance cell is missing in the

literature, though we tried to include the resistance comparison in Fig. 6. Therefore, we inevitably discuss the good performance of the [Ref. 25] cell in terms of power output instead in the revised manuscript as follows:

- line 262 on page 15,

“This power achievement is enhanced significantly compared with data from previously studied BZY-based cells, confirmed in Fig. 1 and supplementary Table 1, and greater than record data from all PCFCs previously developed (650 mW cm^{-2} at $600 \text{ }^\circ\text{C}$)²⁵. ”

- fig 4: why does the OCV of the optimized cell increase only at high H₂ concentrations, in contrast to the other curve? please explain.

: We suspect that this is due to structural characteristics of anode interlayers comprised of multiple layers - supporting anode composite, micro-AFL, and nano-AFL as shown in Fig. 4c - synthesized in varied compositions by multistep post-annealing. Throughout the experiments, we found that integration of the thin and dense BZY membrane onto the porous anode support was extremely challenging and that preparation of a multilayered anode support carefully optimized in terms of composition and geometry was essential. Details of the anode interlayer preparation are described in the supplementary section titled “Optimization of key parameters for microstructural stability and performance improvement: Ni contents and post-annealing temperatures of Ni-BZY nano-AFL”. As discussed in this section, well-ordered nano-size pores are expected to form uniformly in the optimized condition, as shown in supplementary Fig. 5g. This nanoporous structure is favorable for sustaining thin and dense BZY electrolytes and for promoting the charge-transfer reaction at the electrolyte-electrode boundaries. However, we easily anticipated that getting effective gas supply thoroughly through the layers could be challenging through such small pores. Therefore, opening up of these small pores by reduction throughout the anode layers could be retarded significantly, especially when low-concentration hydrogen is used. Moreover, the supply gas should compete against counterflow of the water outgas that is a product of NiO reduction, which implies that hydrogen delivered near the electrolyte could be diluted more. In the case of non-optimized cells, however, the relatively large-scale cracks or spaces between the delaminated layers, as shown in Fig. 4c, throughout the AFLs are expected during the reduction. In addition, the hydrogen supply gas could be delivered more effectively through these large spaces and competition against the counterflow water outgas should be less severe. This is considered the reason the OCV of the non-optimized PCFC increased early with a relatively low concentration of hydrogen (Fig. 4a). This explanation was added to the revised manuscript as shown below:

- line 181 on page 10,

“It should be noted that high OCV was observed in the optimized cell at high concentration of hydrogen (Fig.

4a). We suspect that this is due to the structural characteristics of nano-AFLs, which are comprised of multiple layers with well-ordered nano-size pores, as shown in Fig. 4d. This nanoporous structure is favorable for sustaining thin and dense BZY electrolytes and for promoting the charge-transfer reaction at electrolyte-electrode boundaries. However, it is also anticipated that getting effective gas supply thoroughly through the layers could be challenging through such small pores. Therefore, opening up these small pores by reduction throughout the nano-AFLs could be retarded significantly, especially when low-concentration hydrogen is used. Moreover, the supply gas should compete against the counterflow of the water outgas that is a product of NiO reduction, which implies that hydrogen delivered near the electrolyte could be diluted more. In the case of non-optimized nano-AFLs, however, relatively large-scale cracks or spaces between the delaminated layers form, as shown in Fig. 4c, where the hydrogen supply gas could be delivered more effectively through these large spaces and thus competition against counterflow water outgas should be less severe. For this reason, OCV of the non-optimized PCFC appeared at a relatively early stage with a relatively low concentration of hydrogen, as observed in Fig. 4a.”

- pg 9 line 167: it is clear that electrode delamination decreases performance, but it needs more explanation why it should decrease OCV

: As observed in the low magnitude SEM image of Fig. 4c (left side), the delaminated layers are severely curled-up. In this situation, it is hard not to expect a crack through the membrane. Once a major crack forms at any point of the membrane, there would be crossover of hydrogen through the electrolyte and good OCV would no longer be expected (Fig. 4a). To clarify this aspect related with the poor OCV of the non-optimized cell, the relevant sentence was revised as below:

- line 173 on page 10,

“This delamination is expected to accompany local cracks through the membrane, resulting in the abrupt OCV drop with crossover of hydrogen during the reduction step shown in Fig. 4a.”

- the optimization of the anode and anode functional layer is discussed in great detail (it is an important part as it is the key to fabricating a thin gastight electrolyte on a porous anode substrate). However, also some more details about the PLD process for the electrolyte deposition would be welcome (e.g. how sensitive it is to variations in PLD processing conditions).

: Prior to this work, we conducted optimization of the PLD conditions for fabrication of dense and fully crystallized BZY thin-films of the desired composition, which is already published in a journal [Ref. 49]. The

relationships between film properties and PLD conditions including oxygen background pressure ($p(\text{O}_2)$), the target-to-substrate distance (T-S Dis.), and the substrate temperature were considered critical and have been rigorously discussed in the literature and referenced in this work. In the revision, the previous work was cited additionally in the Methods section with the following sentence, indicating the effectiveness of the PLD conditions used for fabrication of BZY electrolyte.

- line 336 on page 18,

“Validity of this process for growing BZY films is discussed rigorously and confirmed in our previous work⁴⁹.”

- pg 14 line 255 "non proton conducting LSC": I think for LSC there are no data yet, but for other materials there are indications that typical cathode perovskites have a small but nonzero proton solubility and corresponding proton conductivity that expands the reactive zone beyond the triple phase boundary. see e.g. Han (SSI 181 (2010) 1601) and Poetzsch (PCCP 16 (2014) 16446).

: Based on the suggested literature, we confirmed that proton conduction has been observed in LSC, although the proton solubility and corresponding conductivity are extremely low [Ref. 40]. Therefore, the expression “non-proton-conducting LSC” was removed, and the context was changed in the revision as below:

- line 284 on page 15,

“Further improvement is expected by use of a high-performing and stable cathode material substituting for the LSC that has negligible proton conductivity⁴⁰.”

- experimental: please (roughly) indicate the grain sizes of the starting materials. what is the Y-content in the anode layer? what is the lateral cell size? full details for PLD are required (pulse length, repetition frequency, target substrate distance...)

: As suggested, this information is in the revised manuscript as follows:

- line 128 on page 7,

“...exhibiting BZY grain size of approximately 0.5 μm or less and NiO grain size of approximately 2 μm in the sintered body.”

- line 322 on page 17,

“...composition of the anode BZY powder: $\text{BaZr}_{0.85}\text{Y}_{0.15}\text{O}_{3-\delta}$...”

- line 326 on page 18,

“...with a cell size of $1 \times 1 \text{ cm}^2$...”

- line 330 on page 18,

“...and a repetition rate of 10 Hz. The substrate temperature, O_2 background pressure, and target-to-substrate distance were kept at $750 \text{ }^\circ\text{C}$, 6.67 Pa, and 5 cm, respectively, during the deposition.”

- line 340 on page 18,

“...and an area of $0.3 \times 0.3 \text{ cm}^2$...”

Reviewer #3 (Remarks to the Author):

For a paper to be published in a venue as prestigious as Nature Communications it should encompass either a remarkable technological breakthrough, or an exceptional scientific understanding, or ideally, both. This paper sells itself as one in which a technological breakthrough has been achieved on the basis of a scientific understanding and rational solution to an obstacle.

The basic premise of this work is that the high grain boundary impedance has been the cause of poor performance in BZY electrolyte fuel cells, and that the columnar grains of PLD grown BZY films circumvent this problem, leading to exceptional performance. Both the premise (that the grain boundaries have been the culprit) and the final conclusion (that the performance is exceptional) are disputable.

The fuel cell performance reported here by Bae et al. is quite impressive. However, in light of the recent work of Duan et al. (reference 36), it is perhaps less impressive than the authors make claim. As recognized by the present authors, Duan et al. achieved at $600 \text{ }^\circ\text{C}$ a peak power density of 650 mW/cm^2 as compared to the present result of 740 mW/cm^2 . An increase by about 15% over what was a breakthrough result deserves recognition, but within the appropriate context. In Figure 1, in which the context is presented, the authors apparently limit the cells represented to those with an electrolyte of at least 70% Zr on the B site. This limitation makes sense if one wants to try and understand why barium zirconate based cells have hit a

performance limitation, but it does not make sense if one is trying to claim a technological breakthrough. If an electrolyte, such as $\text{Ba}(\text{Ce}_{0.7}\text{Zr}_{0.1}\text{Y}_{0.1}\text{Yb}_{0.1})\text{O}_3 + 1\%\text{NiO}$ as employed by Duan et al., provides attractive performance and greater ease of processability, then the present authors need to explain how their work represents a technological advantage. One point is that BCZYYb is not stable when methane is supplied as the fuel. The present authors have not capitalized on this (likely) advantage of BZY, stability in the presence of carbon-containing fuels. Nor have they explored whether a Ni-BZY containing anode would be immune to carbon deposition when so operated.

The conclusion that grain boundary impedance has limited power output from barium zirconate based fuel cells is based on (1) the observation that no-one has been able to make a good cell from these electrolytes (Figure 1, Table S1), and (2) that the ohmic resistance in cells from the literature which have been characterized by impedance spectroscopy is greater than what one would expect based on the bulk (grain interior) conductivity of the electrolyte material and its thickness, Figure 6(c). This evidence is weak. First, any number of reasons could lead to poor cell performance, so Table S1 alone does not provide evidence of the negative role of grain boundaries. Figure 6(c) is somewhat more convincing, but still weak, as an offset in an impedance spectrum could be due to factors other than grain boundary impedance. As shown by Yamazaki et al. (ref 14) for a typical BZY material with grain size on the order of 0.5 μm , the grain boundary impedance will be about 50% of the total impedance at 600 °C. Therefore, eliminating grain boundaries as done in the present work would effectively double the macroscopic conductivity. This is not enough to explain the poor performance of literature cells, or the high ohmic offsets previously reported. It is also important to recognize that the literature data shown in Figure 6(c) originate from just two laboratories, so any systematic error in those measurements would be exaggerated as a general literature result. Furthermore, because some of the results summarized in Table S1 also reflect PLD grown electrolytes and such growth almost always produced columnar films with few grain boundaries normal to current flow, it is difficult to imagine that those cells suffered from grain boundary impedance. The authors are encouraged to consider other factors that may have led to the high performance observed here. For example, is the stoichiometric control of the electrolyte particularly good? It is known that Ba deficiency can result in low conductivity. Or are the electrode-electrolyte contacts particularly good? It is plausible that PLD growth of the various layers provides better contacts than co-sintering of heterogeneous components.

While I am critical of this work in terms of the interpretation of why the performance is excellent, and I am concerned that there is a bit of exaggeration of the extent to which this is truly a breakthrough, the work is nonetheless impressive and worthy of recognition. I do not believe it is absolutely essential that the authors understand definitively why others have been rather successful in pursuing BZY-based cells in order to proceed to publication. However, a revision must acknowledge that several factors may explain the high performance of their cells relative to others.

: We appreciate your comments, which are critical for improving the quality of our manuscript. Our intention with this paper is to demonstrate the effectiveness of BZY material as an electrolyte for PCFCs or low

temperature ceramic fuel cells by showing high power output using the thin film/anode support structure. It is clear that fuel cell performance is greatly enhanced compared to reference cells with BZY electrolytes, as presented in Table 1 of the supplementary information. We agree that it is insufficient to suggest “single-grain-columnar structure” or grain-boundary-free membrane across the thickness, which is observed in the TEM images, as the sole factor for the performance enhancement. Therefore, we have extended our discussion regarding the reasons for poor performance of the literature BZY cells in comparison with our BZY cell with additional experimental support or reference data as follows.

1. Fabricating thin film BZY electrolytes on a porous anode support

Poor sinterability of BZY has been reported repeatedly [Refs. 6, 9, 15, 19, 18]. The SEM images presented below are from a representative study on this challenge (Nasani *et al.*, Int. J. Hydro. Energy 38 (2013) 8461-8470). The pictures show surface morphologies of BCY, BCZY, and BZY (from left to right) after sintering at 1500 °C for 8 h, with pores clearly formed on the samples made by even such intensive sintering conditions.

The challenge to fabricate BZY films in tightly-integrated grains or large grains has inevitably necessitated the adoption of a relatively thick electrolyte when a porous anode support is used. The thickness of the reference cell BZY electrolyte lies in the range of about 20–30 μm, and this large thickness inevitably increases ohmic resistance and drops power output. In comparison, the thickness of our BZY is ≤2.5 μm and significant reduction in ohmic resistance was expected accordingly. This is discussed in the revised manuscript in the introductory section as shown below.

2. Electrolyte resistance by grain separation

It has been also repeatedly claimed that grain separation is the main culprit for retarded proton transport or large electrolyte resistance in BZY membranes [Refs. 6, 7, 14, 15, 16]. Therefore, minimization, or ideally elimination, of the grain-boundaries is expected to reduce ohmic resistance of BZY electrolytes effectively. Indeed, Yamazaki *et al.* [Ref. 14] demonstrated the conductivity improvement from the large-grained BZY, which is quoted in the reviewer’s comment as well. Thus, adoption of the “single-grain-columnar structure”, free of grain boundary along the membrane thickness, confirmed by the TEM images, should be truly effective in reducing

ohmic resistance and enhancing fuel cell power accordingly.

Regarding the improved fuel cell performance by 1. reduction of membrane thickness and 2. minimization of grain boundary across the membrane, we have revised the discussion in the manuscript as below.

- line 276 in page 15,

“To examine the significant improvement of electrochemical performance, the ohmic and polarization ASRs of representative BZY-PCFCs found in the literature were compared (Fig. 6c,d). An order of magnitude lower ohmic ASRs were achieved in the current work compared to the reference values, as shown in Fig. 6c. This finding can be attributed to the significantly reduced thickness of the BZY electrolyte and, more importantly, the achievement of a corresponding impeding-grain-boundary-free columnar structure. ”

- line 296 on page 16 (figure caption of Fig. 6c),

“**c**, Ohmic area specific resistance estimated from the impedance spectra in **(b)**, compared with the data of representative anode-supported BZY-PCFCs in the literature (1. Xiao *et al.*, 2012⁴¹; 2. Pergolesi *et al.*, 2010⁴²; 3. Luisetto *et al.*, 2012⁴³; 4. Sun *et al.*, 2010⁴⁴; 5. Bi *et al.*, 2011⁴⁵; 6. Bi *et al.*, 2011⁴⁶; 7. Sun *et al.*, 2014⁴⁷; 8. Sun *et al.*, 2013⁴⁸). **d**, Polarization area specific resistance estimated from the impedance spectra in **(b)**, compared with the data of representative anode-supported BZY-PCFCs from the same studies in **(c)**.”

3. Stoichiometry of BZY electrolytes

We appreciate the reviewer’s comments on the BZY stoichiometry issue. It has been well reported that Ba deficiency degrades ion conduction in BZY, as commented. The PLD BZY conditions specified in the Methods section were established throughout the experiments and were confirmed reproducible [Refs. 27, 49]. For confirmation, we prepared a set of BZY films deposited on sapphire substrates under the same PLD conditions used in this work and analyzed their composition by SEM-EDS. The figures below represent the cross-sectional SEM images, the composition data, and the XRD crystallography. The resulting data indicate that our BZY film was deposited in the intended composition of $\text{BaZr}_{0.85}\text{Y}_{0.15}\text{O}_{3-\delta}$ without any secondary phases. From this, we expected no significant flaw in the compositional and crystalline quality of our BZY films that would affect ionic conduction. The figures below are in the revised supplementary information with corresponding explanation in the main article and figure caption as follows:

- line 225 in page 12,

“The X-ray diffraction and SEM-energy dispersive X-ray spectroscopy (EDS) measurement of the BZY electrolyte fabricated using the same PLD conditions on sapphire substrates have confirmed that the stoichiometry matched well to that of one of the PLD targets with no secondary phase, as represented in supplementary Fig. 8.”

- page 16 in the supplementary information (figure caption of supplementary Fig. 8),

“**Supplementary Figure 8. Properties of BZY film fabricated on sapphire substrates under the same PLD conditions as those for fabrication of the BZY electrolyte on the tested PCFCs in the main article: a,** Cross-sectional and surface (embedded) SEM images. **b,** Compositional data measured at three different positions on the surface of the film by using EDS-equipped SEM. **c,** X-ray diffraction data.”

4. Integration of the BZY electrolyte thin film and the porous anode support

Benefits in the adoption of the multiple AFLs fabricated by multistep post-annealing include significantly improved adhesion between the electrolyte film and the anode support. Figs. 4d and 5c clearly confirm well-integrated BZY electrolyte on the AFLs and the anode support. Films deposited in the non-optimized conditions experienced severe delamination (Fig. 4c). In addition, the AFLs with nano-size pores are expected to increase their number of triple phase boundaries (TPBs) where electrolyte, electrode, and space are in physical contact with each other and a majority of charge-transfer reactions are considered to take place. From this structural improvement, enhanced electrode reaction is expected and our EIS confirmed a significantly low polarization ASR (ASR_{pol}) of BZY cells compared to those reported in the literature, as shown in Fig. 6d. We considered that the low ASR_{pol} by improved electrolyte-AFL or electrolyte-anode support integration also contributed significantly to the enhanced fuel cell performance. Additional discussion on ASR_{pol} in relation to fuel cell performance enhancement is included in the revision as follows:

- line 281 on page 15,

“Relatively low polarization ASRs were also observed during the comparison (Fig. 6d). We believe that the nano-size grains of the LSC cathode and the Ni-BZY nano-AFL increased the number of active sites in the electrode reaction. Further improvement is expected by use of a high-performing and stable cathode material substituting for the LSC that has negligible proton conductivity⁴⁰. Moreover, the improved integration of electrolyte and anode support by adoption of the multilayered AFLs using multistep post-annealing has been observed clearly in the cross section of the stack, as presented in Figs. 4d and 5, which is considered to have contributed significantly to the improved charge-transfer reaction, decreased polarization ASRs, and enhanced fuel cell power.”

- line 300 on page 17 (figure caption of Fig. 6d),

“**d**, Polarization area specific resistance estimated from the impedance spectra in **(b)**, compared with the data of representative anode-supported BZY-PCFCs from the same studies in **(c)**.”

The Abstract was also revised reflecting the discussion above as follows:

- line 22 on page 2,

“However, poor sinterability of BZY discourages its fabrication as a thin-film electrolyte and integration on porous anode supports, both of which are considered essential to achieve high power output at LT operation. Here, we fabricated a PCFC using a thin-film-deposited BZY electrolyte with no impeding grain boundaries owing to the columnar structure tightly integrated with nanogranular cathode and nanoporous anode supports, which have exhibited a record-high power output of up to an order of magnitude higher than those of other reported BZY-based PCFCs.”

In addition, the work of Duan et al must be discussed early in the manuscript in terms of context, and the authors need to make a convincing case as to why the BZY cells with complex PLD processing steps would be technologically preferable to BCZYYb. I don't believe a 15% performance gain alone would warrant the processing effort.

: We appreciate the reviewer's valuable comment. As suggested by the reviewer, we have conducted additional experiments to demonstrate the excellent chemical stability of BZY against carbon contamination. The figure below represents the XRD crystallography measured from four types of proton-conducting ceramic powders including $\text{BaCe}_{0.8}\text{Y}_{0.2}\text{O}_{3-\delta}$ (BCY), $\text{BaCe}_{0.7}\text{Zr}_{0.1}\text{Y}_{0.1}\text{Yb}_{0.1}\text{O}_{3-\delta}$ (BCZYYb), $\text{BaCe}_{0.55}\text{Zr}_{0.3}\text{Y}_{0.15}\text{O}_{3-\delta}$ (BCZY), and

BaZr_{0.85}Y_{0.15}O_{3-δ} (BZY) before and after heat treatment at 600 °C for 150 h under 50% CO-50% CO₂ atmosphere. The XRD results clearly confirm excellent stability of BZY against carbon contamination by exhibiting no phase change, whereas other samples have been degraded significantly as indicated by the formation of secondary phases.

Long-term stability under fuel cell operating conditions with a moisturized methanol fuel (70% CH₃OH + 30% H₂O) was also evaluated using an electrolyte-supported BZY cell. The thickness of the BZY electrolyte was about 750 μm, and porous Pt and Pt-Ru layers fabricated by sputtering were used as the cathode and anode, respectively. A BCZYb cell was also prepared for comparison with the same cell configuration to the BZY cell. A constant current density of 3.3 mA cm⁻² was applied to both cells at 450 °C during the test. As shown in the figure below (also included in the supplementary information), excellent stability against carbon impurity is clearly confirmed from the BZY cell over 100 h, whereas fast degradation of the BCZYb was observed with complete dying of the cell after approximately 8 h of operation, though this material is reported to exhibit relatively decent stability against carbon contamination [Yang *et al.*, Science 326 (2009) 126-129]. Duan *et al.* [Ref. 25] also reported that chemical stability of BZY is superior to that of BCZYb from testing with methane fuel, which is specified in the supporting information of this paper.

We believe that these results provide a strong motivation for use of BZY in PCFCs. We have revised the introductory section that emphasized the necessity of BZY materials in terms of chemical stability with additional information regarding XRD and long-term tests in the supplementary information as follows:

- line 44 on page 3,

“This excellent chemical stability of BZY against carbon contamination was also confirmed in our preliminary experiments as described in supplementary Figs. 1 and 2.”

- page 4 in the supplementary information (figure caption of supplementary Fig. 1),

“**Supplementary Figure 1. Chemical stability of representative PC materials.** X-ray diffraction patterns of **a**, $\text{BaCe}_{0.8}\text{Y}_{0.2}\text{O}_{3-\delta}$ (BCY); **b**, $\text{BaCe}_{0.7}\text{Zr}_{0.1}\text{Y}_{0.1}\text{Yb}_{0.1}\text{O}_{3-\delta}$ (BCZYYb); **c**, $\text{BaCe}_{0.55}\text{Zr}_{0.3}\text{Y}_{0.15}\text{O}_{3-\delta}$ (BCZY); and **d**, $\text{BaZr}_{0.85}\text{Y}_{0.15}\text{O}_{3-\delta}$ (BZY) powders before and after exposure to 50% CO-50% CO₂ atmosphere at 600 °C for 150 h.”

- page 5 in the supplementary information (figure caption of supplementary Fig. 2),

“**Supplementary Figure 2. Long-term stability with methanol fuel.** Comparison of long-term stability of $\text{BaZr}_{0.85}\text{Y}_{0.15}\text{O}_{3-\delta}$ (BZY) and $\text{BaCe}_{0.7}\text{Zr}_{0.1}\text{Y}_{0.1}\text{Yb}_{0.1}\text{O}_{3-\delta}$ (BCZYYb) under fuel cell operating conditions of flowing vaporized methanol with water as a fuel. Note that the relatively low power from these cells compared to ones from the anode-support thin film BZY cells of the main article is due to the relatively large thickness of

the electrolyte (about 750 μm) and the use of methanol fuel.”

- page 17 in the supplementary information (added to the Methods section),

“**Chemical stability tests.** For the powder tests, four different PC powders, $\text{BaCe}_{0.8}\text{Y}_{0.2}\text{O}_{3-\delta}$ (BCY), $\text{BaCe}_{0.7}\text{Zr}_{0.1}\text{Y}_{0.1}\text{Yb}_{0.1}\text{O}_{3-\delta}$ (BCZYYb), $\text{BaCe}_{0.55}\text{Zr}_{0.3}\text{Y}_{0.15}\text{O}_{3-\delta}$ (BCZY), and $\text{BaZr}_{0.85}\text{Y}_{0.15}\text{O}_{3-\delta}$ (BZY) were prepared. The solid-state reaction method was used for the powder synthesis of BCY, BCZYYb, and BZY. The raw materials of BaCO_3 , ZrO_2 , Y_2O_3 , and Yb_2O_3 were mixed by using zirconia balls for 24 h, and then calcined at 1300 $^\circ\text{C}$ for 10 h. The calcination was fulfilled repeatedly until a single-phase appeared for each powder. For the BCZY powder, a commercialized powder (K-ceracell Tech.) was used. For evaluation of chemical stability against carbon contamination, the prepared powders were exposed to a flowing gas mixture of CO-CO_2 (50 vol% each) with a flow rate of 200 sccm in a quartz-tube heated at 600 $^\circ\text{C}$ for 150 h. The phase of the powders was examined before and after the heat treatment using X-ray diffraction analysis.

For the fuel cell test, the BZY and the BCZYYb pallet cells were prepared using the synthesized powders pressed at 200 MPa followed by sintering at 1700 $^\circ\text{C}$ and 1500 $^\circ\text{C}$ for 10 h, respectively. The sintered pellets were then grinded to a thickness of 750 μm . Porous Pt and Pt–Ru layers were deposited by radio-frequency sputtering as the cathode and anode, respectively. During the cell tests, vaporized methanol with a water content of 30% was supplied by N_2 carrier gas on the anode side, and the cathode side was open to the air. A constant current density of 3.3 mA cm^{-2} was applied at an operating temperature of 450 $^\circ\text{C}$.”

- page 18 in the supplementary information (added to the characterization in the Methods section),

“In addition, crystallinity and composition data were obtained by using X-ray diffraction (D/MAX-2500, Rigaku) and EDS-equipped SEM, respectively, in the main article.”

REVIEWERS' COMMENTS:

Reviewer #1 (Remarks to the Author):

I think the paper has been revised well with complimentary data, and it can be accepted for publication after some small revisions.

1. Supplementary page 10 line 125: authors should check this sentence. "40-1300 PCFC" should be changed to "40-1200 PCFC".
2. Supplementary page 11 lines 135-140: Except the different morphologies (pore sizes), Ni content factor also should be considered for the differed impedance.

Reviewer #2 (Remarks to the Author):

The authors revised the paper appropriately

Reviewer #3 (Remarks to the Author):

In this revision the authors have addressed several items of concern. In particular, they now properly acknowledge the work of Duan et al, and make clear how BZY based cells are advantageous over those based on mixed zirconate-cerates (as used by Duan et al.).

1. Unfortunately, the authors still argue that the excellent performance of their cells is due to the absence of grain boundaries in the electrolyte. The authors seem to have ignored the reviewer's comments that this cannot be. I repeat:

"As shown by Yamazaki et al. for a typical BZY material with grain size on the order of 0.5 micrometers, the grain boundary impedance will be about 50% of the total impedance at 600C. Therefore, eliminating grain boundaries as done in the present work would effectively double the macroscopic conductivity. This is not enough to explain the poor performance of literature cells, or the high ohmic offsets previously reported." That is to say, the absence of grain boundaries cannot be the main reason for an order of magnitude decrease in ohmic ASR.

The authors need to either come up with an alternative explanation (certainly the fact that the electrolyte is almost an order of magnitude thinner than typical electrolytes is a good candidate!), or state that they don't know. In the response to Reviewer 1, many factors that contribute to the ohmic offset are correctly identified, but somehow missing from the manuscript. The authors cannot put forth in the manuscript an explanation that is not supported by analysis, and in fact contradicts what is known about the behavior of this electrolyte material.

2. The authors argue that NiO that has been used by others as a sintering aid is detrimental to the electrolyte properties. It is inconceivable that a 2 micrometer thick electrolyte supported on a Ni-BZY anode would remain free of Ni during long-term operation. So, if NiO as a sintering aid were truly problematic, the present work would eventually suffer the same fate. In fact, NiO has only minor negative impact on electrolyte conductivity and the statements suggesting it is a major impediment should be removed.

3. The other reviewers have remarked on the OCV. I agree with those concerns. It is typical to observe an OCV that increases with decreasing temperature (both because the Gibbs energy for water formation increases with decreasing temperature and electronic leakage through the electrolyte decreases). Here, the value is oddly almost exactly 1.0 V at all temperatures. The revision

“For all the measured temperatures, high OCVs (over 1 V) were observed, implying proper electronic insulation and gas-tightness of the thin BZY electrolyte.”

is somewhat of an exaggeration as the data seem to show that the OCV is 1.0, not over 1 V. Instead of acknowledging the unchanging OCV as a liability, the authors present it as a strength. I agree that the OCV is good for such a thin cell, but the strangely temperature independent behavior should be at least acknowledged. A 20 mV variation is quite easy to detect, and instrumental insensitivity to 20 mV cannot be the reason that the OCV appears temperature insensitive.

4. Reviewer 1 (item 2) asks about the relationship between the ASR and the peak power densities. The reviewer apparently estimated peak power densities assuming linear I-V curves. Figure 6a clearly shows these curves are non-linear, particularly at low temperature. The authors should comment on this non-linearity. The behavior indicates that at low temperature, poor electrocatalysis dominates the IV characteristics (steep slope at low current), whereas at high temperature, mass diffusion losses become important (downward curvature at high current). The appearance of mass diffusion loss as a power limiting factor is not surprising given the small pore sizes in the anode, a point that the authors discuss in the rebuttal and should be addressed in the text.

The authors would like to thank the editor and reviewers for giving us the opportunity to submit a revised manuscript. Revisions were carried out according to the reviewers' suggestions. The reviewer comments are shown in black and our responses to the comments are in blue. The manuscript revisions are marked in red.

REVIEWERS' COMMENTS:

Reviewer #1 (Remarks to the Author):

I think the paper has been revised well with complimentary data, and it can be accepted for publication after some small revisions.

1. Supplementary page 10 line 125: authors should check this sentence. "40-1300 PCFC" should be changed to "40-1200 PCFC".

: This part has been confirmed and corrected according to the comment.

2. Supplementary page 11 lines 135-140: Except the different morphologies (pore sizes), Ni content factor also should be considered for the differed impedance.

: Previous studies have shown that a Ni content change of about 10% has a very limited effect of less than 10% on the polarization resistance (L. Bi et al. *J Electrochem Soc* 158 (7) B797-B803 (2011)). Based on these results, it can be concluded that the remarkably different polarization resistances identified in the comparison of 40-1200 PCFC and 50-1300 PCFC are mainly due to the difference in the microstructure of the nano AFL. This discussion has been included in the supplementary information as well as the literature referred to below.

- page S12 in Supplementary Information,

“Changes in the Ni content in the composition range of 40–50 vol% may have some effect on the polarization resistance, but any change in the values is reported to have only minor impact on the electrode performance³². This observation supports that the main cause of the extremely reduced polarization impedance, seen for 50-1300 PCFC in supplementary Fig. 7b, is improvement in the microstructure rather than increase in the Ni content.”

Reviewer #2 (Remarks to the Author):

the authors revised the paper appropriately

Reviewer #3 (Remarks to the Author):

In this revision the authors have addressed several items of concern. In particular, they now properly acknowledge the work of Duan et al, and make clear how BZY based cells are advantageous over those based on mixed zirconate-cerates (as used by Duan et al.).

1. Unfortunately, the authors still argue that the excellent performance of their cells is due to the absence of grain boundaries in the electrolyte. The authors seem to have ignored the reviewer’s comments that this cannot be. I repeat:

“As shown by Yamazaki et al. for a typical BZY material with grain size on the order of 0.5 micrometers, the grain boundary impedance will be about 50% of the total impedance at 600C. Therefore, eliminating grain boundaries as done in the present work would effectively double the macroscopic conductivity. This is not enough to explain the poor performance of literature cells, or the high ohmic offsets previously reported.” That is to say, the absence of grain boundaries cannot be the main reason for an order of magnitude decrease in ohmic ASR.

The authors need to either come up with an alternative explanation (certainly the fact that the electrolyte is almost an order of magnitude thinner than typical electrolytes is a good candidate!), or state that they don’t know. In the response to Reviewer 1, many factors that contribute to the ohmic offset are correctly identified, but somehow missing from the manuscript. The authors cannot put forth in the manuscript an explanation that is not supported by analysis, and in fact contradicts what is known about the behavior of this electrolyte material.

: We have revised the manuscript regarding the reduction of ohmic ASR and cell performance as follows, in accordance with the reviewers' opinion. In addition, as described in the response to reviewer 1, the improved

adhesion of the BZY electrolyte on the anode is believed to have contributed to the ohmic ASR reduction.

- line 47 on page 3,

“The reported poor performance of BZY-PCFCs is mainly due to high ohmic resistance of the electrolyte. One probable contributor is the highly resistive grain-boundaries of BZY in proton conduction, resulting in large ohmic resistance and low power outputs of the PCFC,”

- line 62 on page 4,

“The most straightforward approach to lowering the ohmic resistance of the BZY electrolyte is to reduce its thickness while eliminating the impeding grain boundaries.”

- line 247 on page 11,

“These results suggest that the significantly reduced thickness of the BZY electrolyte is the main cause of the improved cell performance. The improvement in bonding between the porous anode and the thin and dense columnar BZY layer, as shown in Fig. 5, also seems to have contributed to the reduction in ohmic ASRs.”

- line 266 on page 12,

“On the anodes, a grain-boundary-free columnar BZY electrolyte with significantly reduced thickness was successfully fabricated by PLD. This thin BZY electrolyte is believed to substantially reduce the ohmic resistance compared with those of BZY-PCFCs quoted in literature, which is the main reason for the cell performance enhancement.”

2. The authors argue that NiO that has been used by others as a sintering aid is detrimental to the electrolyte properties. It is inconceivable that a 2 micrometer thick electrolyte supported on a Ni-BZY anode would remain free of Ni during long-term operation. So, if NiO as a sintering aid were truly problematic, the present work would eventually suffer the same fate. In fact, NiO has only minor negative impact on electrolyte conductivity and the statements suggesting it is a major impediment should be removed.

: The influence of the addition of NiO on the electrolyte conductivity was confirmed minor through reviews of related literatures. Upon acceptance of the reviewer's opinion, the following statement was amended.

- line 57 on page 4,

“Solid-state reactive sintering, where material synthesis and sintering are carried out simultaneously using nano-size precursors, has enabled the growth of relatively large BZY grains and effectively reduced grain-boundary resistance^{14,21}. However, a fuel cell having a BZY electrolyte with such large grain sizes (~1 μm) has not been reported yet.”

3. The other reviewers have remarked on the OCV. I agree with those concerns. It is typical to observe an OCV that increases with decreasing temperature (both because the Gibbs energy for water formation increases with decreasing temperature and electronic leakage through the electrolyte decreases). Here, the value is oddly almost exactly 1.0 V at all temperatures. The revision

“For all the measured temperatures, high OCVs (over 1 V) were observed, implying proper electronic insulation and gas-tightness of the thin BZY electrolyte.”

is somewhat of an exaggeration as the data seem to show that the OCV is 1.0, not over 1 V. Instead of acknowledging the unchanging OCV as a liability, the authors present it as a strength. I agree that the OCV is good for such a thin cell, but the strangely temperature independent behavior should be at least acknowledged. A 20 mV variation is quite easy to detect, and instrumental insensitivity to 20 mV cannot be the reason that the OCV appears temperature insensitive.

: As the reviewer noted, the OCV value is indicated to be about 1 V. We observed an increase in the OCV (~10 mV) then the temperature was reduced to 450 °C, but the change is rather insensitive to temperature, as the reviewer mentioned. Sealing issues, which we experienced in other experiments, can be a possible cause. Therefore, the relevant part is revised as follows:

- line 230 on page 11,

“The OCV values were about 1.0 V, which can be considered to be in a reasonable range compared to that of the previously reported BZY-based PCFC⁴¹⁻⁴⁸, especially considering the low thickness of the electrolyte. It implies that the thin BZY electrolyte has the appropriate structural integrity to function as an electrolyte. However, the OCV is rather insensitive to temperature change, which may originate from certain leakage issues such as sealing.”

4. Reviewer 1 (item 2) asks about the relationship between the ASR and the peak power densities. The reviewer

apparently estimated peak power densities assuming linear I-V curves. Figure 6a clearly shows these curves are non-linear, particularly at low temperature. The authors should comment on this non-linearity. The behavior indicates that at low temperature, poor electrocatalysis dominates the IV characteristics (steep slope at low current), whereas at high temperature, mass diffusion losses become important (downward curvature at high current). The appearance of mass diffusion loss as a power limiting factor is not surprising given the small pore sizes in the anode, a point that the authors discuss in the rebuttal and should be addressed in the text.

: We discussed the non-linearity of the I-V curve as follows, according to the reviewer's suggestion.

- line 219 on page 10,

“In Fig. 6a, a drop in the voltage at a low current is observed at less than 500 °C, whereas a fall curve at a higher current is observed at 600 °C. This is because the electrode response is limited to other factors at different temperatures. Specifically, charge transfer reactions are considered to dominate overall electrode kinetics at low temperatures. A temperature increase to 600 °C is expected to help improve the rate of electrochemical reactions and mass diffusion can dominate the electrode process because the reactants can still undergo transfer through small pores present in the nano-AFL.”

* Regarding the SEM images used in response to the first comment of Reviewer #3 in the previous response letter, we have obtained the permission for use of the image from the journal (Nasani *et al.*, *Int. J. Hydro. Energy* 38 (2013) 8461-8470. License Number: 4014080939522).